# AdS-GNN - a Conformally Equivariant Graph Neural Network

**Maksim Zhdanov**[*]
AMLab, University of Amsterdam
m.zhdanov@uva.nl

**Nabil Iqbal**[*]
Department of Mathematical Sciences, Durham University
AMLab, University of Amsterdam
nabil.iqbal@durham.ac.uk

**Erik Bekkers**
AMLab, University of Amsterdam
e.j.bekkers@uva.nl

**Patrick Forré**
AMLab, AI4Science Lab, University of Amsterdam
p.d.forre@uva.nl

## Abstract

Conformal symmetries, i.e. coordinate transformations that preserve angles, play a key role in many fields, including physics, mathematics, computer vision and (geometric) machine learning. Here we build a neural network that is equivariant under general conformal transformations. To achieve this, we lift data from flat Euclidean space to Anti de Sitter (AdS) space. This allows us to exploit a known correspondence between conformal transformations of flat space and isometric transformations on the AdS space. We then build upon the fact that such isometric transformations have been extensively studied on general geometries in the geometric deep learning literature. We employ message-passing layers conditioned on the proper distance, yielding a computationally efficient framework. We validate our model on tasks from computer vision and statistical physics, demonstrating strong performance, improved generalization capacities, and the ability to extract conformal data such as scaling dimensions from the trained network.

## 1 Introduction

The notion of *symmetry* is a key tool both in our understanding of nature and for the construction of machine learning systems that perceive nature. The construction of *equivariant* neural network architectures that encode specific symmetries has powerful advantages both conceptual and computational. In particular, much work has been dedicated to building networks that are equivariant under familiar symmetries such as rotations and translations.

In this work, we will study the symmetry group of *conformal transformations* i.e. the set of transformations on $\mathbb{R}^d$ that preserve angles. This includes translations, rotations, reflections, scalings and so called special conformal transformations. Scale transformations – i.e. rigid rescalings of the whole system – are an extremely important subgroup of the conformal group. Conformal and scale invariance play a central role in many diverse fields. To give some examples: biological visual systems seem to exhibit insensitivity to scale (Logothetis et al., 1995; Han et al., 2020). Physical systems undergoing a second-order phase transition have fluctuations at all scales; they are generally conformally invariant at the critical point (Cardy, 1996), and can usefully be described by *conformal field theories*

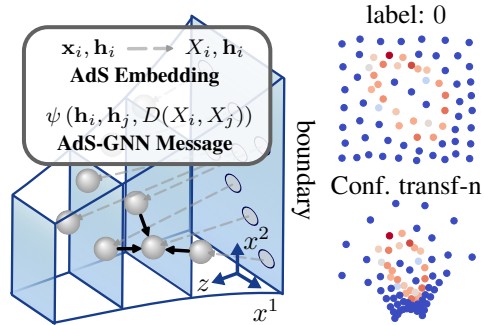

Figure 1: AdS-GNN lifts points from Euclidean space to Anti de Sitter space and computes message passing conditioned on the proper distance.

---

[*]Equal contribution.

(Di Francesco et al., 1997). Indeed in physics most systems exhibiting scale invariance also exhibit conformal invariance "for free" (Polchinski, 1988; Nakayama, 2015). Diverse applications also exist in computational geometry and computer vision (see e.g. (Sharon & Mumford, 2006; Lei et al., 2023))

In this work, we construct a neural network that is equivariant under conformal transformations, motivated by applications to computer vision and to physical systems. Our approach acts on point clouds and lifts the data into an auxiliary higher dimensional space called Anti de Sitter (AdS) space. As outlined below, this approach is inspired by ideas in conformal field theory in theoretical physics. We validate our construction on some tasks drawn from computer vision and statistical physics.

## 2 Previous work

Construction of equivariant neural networks under general symmetry groups began with (Cohen & Welling, 2016) and has since evolved into a well-studied field (see, e.g., (Bronstein et al., 2021; Weiler et al., 2023) for reviews). Advancements include treatments of isometric transformations on general Riemannian manifolds (Weiler et al., 2023) and extensions to semi-Riemannian manifolds (Zhdanov et al., 2023; 2024). The significant benefits of incorporating equivariance, particularly for point cloud data and even at large computational scales, are increasingly demonstrated (Vadgama et al., 2025; Brehmer et al., 2024), highlighting advantages in performance and efficiency. Lie group theory has been employed to develop equivariant networks for broad classes of transformations, such as general affine transformations (Mironenco & Forré, 2024), or to leverage specific algebraic structures like adjoint actions on Lie algebras (Lin et al., 2024). However, despite progress in handling Euclidean symmetries, scale equivariance, or even these more general Lie groups, these methods do not address the full conformal group, which uniquely includes non-affine special conformal transformations.

The pursuit of scale equivariance, a critical component of conformal symmetry, is the closest related area to our work on full conformal equivariance, with various approaches developed from specialized convolutional architectures (Bekkers, 2020; Sosnovik et al., 2019) to techniques like Fourier layers for robust scale handling (Rahman & Yeh, 2023). Our use of AdS space connects conceptually to *scale space* theories (Witkin, 1987; Worrall & Welling, 2019), where the extra dimension corresponds to scale. However, a crucial distinction is that the geometry of AdS space enforces equivariance under the larger group of all conformal transformations by construction, not just scale or isometric transformations. This approach also finds resonance in physics with recent explorations of using neural networks to model aspects of conformal field theory (Halverson et al., 2024).

## 3 Conformal symmetry and Anti de Sitter space

### 3.1 Conformal transformations

A detailed and self-contained account of conformal transformations is given in Appendix D; here we present a brief review. Formally, a *global conformal transformation* of the Euclidean space $\mathbb{R}^d$ is an injective smooth map $\varphi : \mathbb{R}^d \setminus \{x_\varphi\} \to \mathbb{R}^d$, $x \mapsto \varphi(x)$, defined on $\mathbb{R}^d$ except on a possible point $x_\varphi$ [1] such that local angles are preserved, i.e. for all $x \in \mathbb{R}^d \setminus \{x_\varphi\}$ and $v_1, v_2 \in \mathbb{R}^d \setminus \{0\}$ we require[2]:

$$\frac{\langle \varphi'(x)v_1, \varphi'(x)v_2 \rangle}{\|\varphi'(x)v_1\| \cdot \|\varphi'(x)v_2\|} =: \cos(\angle(\varphi'(x)v_1, \varphi'(x)v_2)) \stackrel{!}{=} \cos(\angle(v_1, v_2)) := \frac{\langle v_1, v_2 \rangle}{\|v_1\| \cdot \|v_2\|}, \quad (1)$$

where $\langle ., . \rangle$ and $\|.\|$ denote the standard Euclidean scalar product and norm, resp., and $\varphi'(x)$ the Jacobian matrix of $\varphi$ at $x$. Note, that, in contrast to *isometric transformations*, we do not require that the distances/norms are preserved.

The *group of all global conformal transformations* of $\mathbb{R}^d$ is denoted by $\mathrm{Conf}_g(\mathbb{R}^d)$. We also define the *(restricted) conformal group* $\mathrm{Conf}(\mathbb{R}^d)$ as the connected component of the identity of $\mathrm{Conf}_g(\mathbb{R}^d)$.

---

[1] By allowing $\varphi$ to not be defined on a certain point $x_\varphi \in \mathbb{R}^d$, we effectively allow $\varphi$ to map $x_\varphi$ to the "points at infinity" $\infty$ in the conformal compactification $\mathbb{S}^d$ of $\mathbb{R}^d$. In fact, every *global* conformal transformation of $\mathbb{R}^d$ uniquely extends to an angle-preserving diffeomorphism of $\mathbb{S}^d$ for $d \geq 2$, see (Schottenloher, 2008) Thm. 2.6-2.11, or D.4.2 and D.4.3. This is why we introduce the definition with $x_\phi$ in this way.

[2] An equivalent, but more general and abstract, definition is provided in Definition D.1.2 and Lemma D.1.1.

It is shown [3] that for $d \geq 2$ the global conformal group $\mathrm{Conf}_{\mathrm{g}}(\mathbb{R}^d)$ is isomorphic to the *projective orthogonal group*:

$$\mathrm{PO}(d+1,1) := \mathrm{O}(d+1,1)/\{\pm 1\},$$

where the role of the quotient is described in Appendix D in Proposition D.4.1.

It is instructive to consider the action of the group on points in terms of separated parameters. In particular, a general element $G$ of the group can be written in terms of parameters $(\lambda, t, b, M) \in \mathbb{R}_{>0} \times \mathbb{R}^d \times \mathbb{R}^d \times \mathrm{O}(d)$ and acts on a point $x \in \mathbb{R}^d$ through the composition of maps:

$$x' = x + t \qquad x' = Mx \qquad x' = \lambda x \qquad \frac{x'}{\|x'\|^2} = \frac{x}{\|x\|^2} - b, \qquad (2)$$

| translations | rotations | scalings | special conformal transformations |

resulting in a transformed point $x' = Gx$. These $(d+1)(d+2)/2$ parameters then assemble into an element of $\mathrm{O}(d+1,1)$, see e.g. (Di Francesco et al., 1997) for a review.

Above we have explained how points $x \in \mathbb{R}^d$ transform under the global conformal group. In a typical application we will often be dealing with *conformal fields* $\phi(x)$ defined on this space. The transformation of these fields is governed by the representation theory of the global conformal group[4], which is non-trivial and we do not review it here, except to state that a privileged role is played by a basis of fields called *conformal primaries*, which we denote by $\mathbb{O}(x)$. Associated with each primary field is a number $\Delta_{\mathbb{O}}$ called the *conformal dimension* which characterizes the field in question[5]. Conformal primaries are distinguished by the fact that they transform as simply as possible under the group, and in particular under scaling of the coordinate they transform by a multiplicative factor:

$$\mathbb{O}'(x') := \mathbb{O}'(\lambda x) = \lambda^{-\Delta_{\mathbb{O}}} \mathbb{O}(x) \qquad (3)$$

**Example.** *A simple example is the electric field in ordinary electrodynamics; the usual inverse square law near a point charge $\vec{E}(x) \sim \frac{\hat{x}}{|x|^2}$ is consistent with equation 3 if $\Delta_E = 2$. Another example is image data viewed as a scalar field $p(x)$ denoting the pixel value; under a conformal transformation only the argument of the field changes and not its value, and thus we would take $\Delta_{\mathrm{pixel}} = 0$.*

In physical applications the values of $\Delta_{\mathbb{O}}$ for a given conformally invariant system – e.g. a particular kind of phase transition – are often of great interest[6], as they are pure numbers that are *universal*: they generally do not depend on the microscopic details of the system, only on the symmetries and the spatial dimensionality.

## 3.2 THE ANTI DE SITTER SPACE $\mathrm{AdS}^{d+1}$

To organize the data, we will instead lift it from $\mathbb{R}^d$ to an auxiliary space with *one higher dimension*, i.e. *Anti de Sitter space* $\mathrm{AdS}^{d+1}$. This space can conveniently be understood in terms of a submanifold of $\mathbb{R}^{d+1,1}$, equipped with its natural metric

$$\eta = \mathrm{diag}(-1, 1, 1 \cdots, 1) \qquad (4)$$

Consider the submanifold given by the constraint $\|Y\| = -1$, where $Y \in \mathbb{R}^{d+1,1}$. As shown in Figure 2, this submanifold has two connected components, each of which is defined to be a copy of

---

[3]see Theorem D.4.2, Appendix D, Proposition D.4.1 and (Schottenloher, 2008) Thm. 2.6-2.11 for details.

[4]Elementary reviews can be found in (Simmons-Duffin, 2017; Di Francesco et al., 1997).

[5]It may be helpful to consider fields transforming under the rotation group $\mathrm{SO}(3)$: there are scalar fields, spinor fields, vector fields, etc. which are characterized by a discrete parameter called the *spin* $s$ which takes discrete values $s \in \frac{\mathbb{Z}}{2}$, i.e. $s_{\mathrm{scalar}} = 0, s_{\mathrm{vector}} = 1$, etc. $\Delta$ can very loosely be thought of as the analogue of "spin" for scale transformations. The fact that it is now a continuous variable is related to the fact that the conformal group is non-compact.

[6]See e.g. the 1982 Nobel Prize lecture by (Wilson, 1983) for a historical overview, or (Cardy, 1996; Kardar, 2007; Zinn-Justin, 2021) for textbook treatments.

$\text{AdS}^{d+1}$.[7] For concreteness, we will present all formulas working with the "top" branch, i.e. the one which has $Y^0 > 0$, and we define:

$$\text{AdS}^{d+1} := \{Y \in \mathbb{R}^{d+1,1} \mid \|Y\| = -1, Y^0 > 0\}. \tag{5}$$

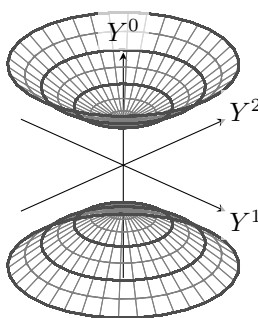

Figure 2: Example of hyperboloid $\|Y\| = -1$ embedded in $\mathbb{R}^{2,1}$, constituting two copies of $\text{AdS}^2$.

There is a natural action of $\text{PO}(d + 1, 1) = \text{O}(d + 1, 1)/\{\pm 1\}$ on this submanifold, given by $\text{O}(d+1, 1)$ rotations on $Y$ on each connected component. This forms the *isometry group* $\text{Isom}(\text{AdS}^{d+1})$ of $\text{AdS}^{d+1}$, see (McKay, 2023) Theorem 7.5 or D.2.6. As mentioned before, note that $\text{PO}(d + 1, 1)$ is also precisely the global conformal group $\text{Conf}_g(\mathbb{R}^d)$ of $\mathbb{R}^d$. This is not an accident: the fact that local operations in the interior of $\text{AdS}^{d+1}$ result in conformally invariant operations on $\mathbb{R}^d$ is very well known in the context of the AdS/CFT correspondence in quantum gravity (Maldacena, 1999; Gubser et al., 1998; Witten, 1998), which states that under some circumstances a quantum gravity theory on $\text{AdS}^{d+1}$ is equivalent to a quantum field theory with conformal symmetry on $\mathbb{R}^d$. Also, see Remark D.2.7.

Here we will not use any of the dynamics of quantum gravity, but we will exploit some of the well-studied kinematics of that correspondence as a convenient tool to build convolutional kernels on $\text{AdS}^{d+1}$. Indeed, a general framework for constructing convolutional layers that are equivariant under isometry groups of any pseudo-Riemannian manifold was given in (Weiler et al., 2023; Zhdanov et al., 2024), and our work can be viewed as building on a special case of that.

To be more precise, we first place explicit coordinates $X = (X^1, \ldots, X^{d+1})$ on $\text{AdS}^{d+1}$. It will be convenient to separate these coordinates as $X = (X^1, \ldots, X^{d+1}) = (x^1, \ldots, x^d, z) = (x, z) \in \mathbb{R}^d \times \mathbb{R}_{>0}$ where $x \in \mathbb{R}^d$ and $z \in \mathbb{R}_{>0}$ is an extra dimension. We can now solve the hyperboloid constraint equation 5 in terms of these coordinates as

$$Y^0 := \frac{z^2 + \|x\|^2 + 1}{2z}, \qquad Y^a := \frac{x^a}{z}, \qquad Y^{d+1} := \frac{z^2 + \|x\|^2 - 1}{2z} \tag{6}$$

The Riemannian metric on $\text{AdS}^{d+1}$ is the induced metric on this hyperboloid[8].

$$ds^2 = \sum_{\mu,\nu=1}^{d+1} g_{\mu\nu}(X)dX^\mu dX^\nu = \frac{1}{z^2}\left(\sum_{a=1}^{d}(dx^a)^2 + dz^2\right). \tag{7}$$

Finally, we record how the isometry group acts on $\text{AdS}^{d+1}$ as $X' = GX$. Using the same parameters as in equation 2 we have:

$$(x', z') = (x + t, z) \qquad (x', z') = (Mx, z) \qquad (x', z') = (\lambda x, \lambda z), \tag{8}$$

$$\left(\frac{x'}{\|x'\|^2 + z'^2}, \frac{z'}{\|x'\|^2 + z'^2}\right) = \left(\frac{x}{\|x\|^2 + z^2} + b, \frac{z}{\|x\|^2 + z^2}\right). \tag{9}$$

A key point is that the manifold (7) has a $d$-dimensional boundary at $z = 0$. This boundary is mapped to itself under the isometries. Furthermore, the isometry group acts on the boundary points $(x^a, z = 0)$ precisely as in (2). Thus one should imagine that conformal data on $\mathbb{R}^d$ "lives on the boundary of $\text{AdS}^{d+1}$". In what follows a key role will be played by the $\text{PO}(d + 1, 1)$-invariant proper distance between $D(X, X')$ between two points in $\text{AdS}^{d+1}$, which is related to the absolute inner product $|\langle Y, Y'\rangle|$ and is given explicitly by:

$$\cosh D(X, X') := |\langle Y, Y'\rangle| \overset{!}{=} \frac{z^2 + z'^2 + \sum_{a=1}^{d}(x^a - x'^a)^2}{2zz'}. \tag{10}$$

---

[7]For a more abstract and general definition of AdS for general quadratic spaces we refer to Definition D.2.3 and the corresponding (partial) parameterization presented here to Appendix D.6.

[8]For the details, see Proposition D.6.1.

## 4 ADS-GNN

We will now describe how to use these ideas to formulate a conformally equivariant neural network by extending the data from the boundary into the bulk of $\text{AdS}^{d+1}$.

---

**Algorithm 1** AdS Embedding

---

**Require:** $X = \{\mathbf{x}_i\}_{i=1}^N \subset \mathbb{R}^d$, $k_{\text{lift}} \in \mathbb{N}$, $z_0 \in \mathbb{R}$
1: **for** each point $i \in \{1, \dots, N\}$ **do**
2: $\quad$ $z_i \leftarrow z_0$
3: $\quad$ $\text{neighbors}_i \leftarrow \text{KNN}(x_i, X, k_{\text{lift}})$
4: $\quad$ $(\hat{x}_i, \hat{z}_i) \leftarrow \text{ComputeAdSCoM}(\text{neighbors}_i)$
5: **end for**
6: **return** $\{(x_i, \hat{z}_i)\}_{i=1}^N \subset \text{AdS}^{d+1}$

---

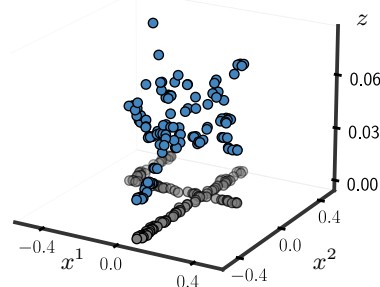

### 4.1 EMBEDDING POINTS IN AdS

Consider a point cloud of $N$ points in $\mathbb{R}^d$, $\{x_i\}$, $i \in \{1, \cdots, N\}$. We would now like to lift this data into the bulk of $\text{AdS}^{d+1}$ in a manner that preserves the symmetries.

A first attempt from the correspondence of symmetries shown in (8) is to simply embed each of the points directly into the boundary $z = 0$, i.e. with $X_i^\mu = (x_i^a, z = 0)$. However the metric (7) has a singularity at $z = 0$ – e.g. from (10) note that each point at $z = 0$ is at infinite proper AdS distance from any points at $z > 0$ – and thus such an attempt will require us to pick a regulating value of $z$. Using a fixed constant will explicitly break the symmetries.

We instead determine a value of $z_i$ for each point using an approach outlined in Algorithm 1. We first embed each point into AdS using

$$X_i^\mu = (x_i^a, z = z_0) \tag{11}$$

with a small regulator $z_0$. For each point, we then compute the *AdS center of mass* $\hat{X} = (\hat{x}_i, \hat{z}_i)$ of its $k_{\text{lift}}$ nearest neighbours. The AdS center of mass is a generalization to hyperbolic space of the familiar notion of the center of mass from flat Euclidean space. It can be computed easily using an approach due to (Galperin, 1993) which we review in Appendix A.2.

The geometry of AdS implies that the center of mass will generally be deeper inside than the original points. Importantly the $z$ value of the centroid now has a finite limit as $z_0 \to 0$, in which case it depends on the (appropriately averaged) relative separation of the points. We then perform a final embedding of the point using this $z$ value, i.e.

$$X_i^\mu = (x_i^a, \hat{z}_i) . \tag{12}$$

Intuitively, the $z$ coordinate corresponds to the *length scale* of the degrees of freedom of a system [9]. Our choice above amounts to saying that the appropriate length scale for a point $x_i$ is related to its distance from its neighbours. This exactly preserves scale invariance, but it gently breaks special conformal transformations. This is expected on physical grounds, as generally any choice of regulator necessarily breaks conformal invariance (Cardy, 1996). In experiments, we check generalization under special conformal transformations empirically and verify that the breaking is mild.

**Proposition 4.1.1.** *The lifting procedure (i.e. the map from $x^a \in \mathbb{R}^d$ to $X^\mu \in \text{AdS}^{d+1}$ described in equation 12) is equivariant under the subgroup of the conformal group generated by translations, rotations and dilatations.*

This is proved in Appendix A.3. Finally, we have discussed lifting the points $x_i$. The input data may also have some *features* $h_i^{\text{input}}$ associated to each point $x_i$. They should be interpreted as a sample of an underlying conformal field $\mathbb{O}(x)$ with dimension $\Delta$, where $h_i^{\text{input}} = \mathbb{O}(x_i)$. This boundary feature should be contrasted with the feature associated with a bulk point in $\text{AdS}^{d+1}$; this is a scalar

---

[9]This is familiar from the physics of the AdS/CFT correspondence, where it is well-understood that the infrared physics lives deeper in the bulk (Susskind & Witten, 1998).

and does not transform with an associated factor of $\lambda^\Delta$ as in equation 3. Said differently, the full dependence of bulk features under scaling arises from its dependence on an extra coordinate $z$. This difference in representation is taken into account by lifting the feature as follows

$$h_i^{\text{lifted}} = \hat{z}_i^\Delta h_i^{\text{input}} \tag{13}$$

For many cases (e.g. image data) the input feature will have $\Delta = 0$ and this step may be skipped. This relation has an analogue in terms of bulk-to-boundary propagators from the AdS/CFT correspondence, where such factors of $z^\Delta$ relate physics in the bulk (i.e. $h_i^{\text{lifted}}$) to that of the boundary (i.e. $h_i^{\text{input}}$)[10]

Given this set of points $\{X_i\}$ in $\text{AdS}$, we now operate on it using a graph neural network.

## 4.2 INVARIANT MESSAGE PASSING

We begin by studying scalar features. To orient ourselves, we recall first an earlier model, that of E(n) Equivariant Graph Neural Networks (EGNNs) (Satorras et al., 2021; Liu et al., 2024). These are graph neural networks that are equivariant to flat-space rotations, translations, reflections and permutations. The input to the model is a graph $G = (\mathcal{V}, \mathcal{E})$ whose vertices are embedded into Euclidean space $\mathbb{R}^d$. We denote the position of node $v_i$ as $\mathbf{p}_i \in \mathbb{R}^d$ and its latent $D$-dimensional feature vector of node $v_i$ as $\mathbf{h}_i$. The $l$-th layer of EGNN is then defined as

$$\mathbf{m}_{ij} = \psi_e(\mathbf{h}_i^l, \mathbf{h}_j^l, \|\mathbf{p}_i - \mathbf{p}_j\|_2), \qquad \text{EGNN message} \tag{14}$$

$$\mathbf{h}_i^{l+1} = \psi_h(\mathbf{h}_i^l, \mathbf{m}_i), \quad \mathbf{m}_i = \sum_{j \in \mathcal{N}(i)} \mathbf{m}_{ij}, \qquad \text{aggregate + update}$$

where here $\mathcal{N}(i)$ represents the set of neighbours of node $v_i$, $\psi_e, \psi_h$ are message and update MLPs, which we see depend only on the pairwise distance in $\mathbb{R}^d$ between nodes.

We adopt the model above to operate on $\text{AdS}$ where a graph $G$ is embedded. As above, each node $v_i$ has a position $X_i \in \text{AdS}^{d+1}$ and a latent feature vector $\mathbf{h_i}$. If edges $\mathcal{E}$ are not provided in the graph, we induce connectivity with $k_{\text{con}}$ nearest neighbours using the $\text{AdS}$ proper distance (10).

In the message function (14), we also use the $\text{AdS}$ proper distance instead of the Euclidean one, i.e.

$$\mathbf{m}_{ij} = \psi_e(\mathbf{h}_i^l, \mathbf{h}_j^l, D(X_i, X_j)), \qquad \textbf{AdS-GNN } \text{message} \tag{15}$$

which yields an efficient conformal group equivariant GNN without substantial computational overhead compared to its Euclidean counterpart. Note that by making use of the $\text{AdS}$ proper distance, we introduce a notion of locality both in ordinary space and in scale (as represented by the $z$-coordinate). Although the embedding of the point cloud mildly breaks special conformal transformations, the graph neural network itself is exactly invariant under all of $\text{Conf}(\mathbb{R}^d)$.

The message function in Eq. 15 only depends on distance and therefore is restricted to invariant features. To enable more expressive equivariance, we want to exploit the fact that $\text{AdS}^{d+1}$ is a $\text{Conf}(\mathbb{R}^d)$-equivariantly embedded sub-manifold of $\mathbb{R}^{d+1,1}$, as discussed in Section A.1. This allows us to employ $O(d + 1, 1)$-equivariant neural networks from (Ruhe et al., 2023), which operate on multivectors, i.e. elements of the Clifford Algebra $\text{Cl}(\mathbb{R}^{d+1,1})$. This yields the following message structure:

$$\mathbf{M}_{ij} = \psi_e(\mathbf{H}_i^l, \mathbf{H}_j^l, X_i, X_j), \qquad \textbf{AdS-CEGNN } \text{message} \tag{16}$$

where $\mathbf{M}_{ij}, \mathbf{H_i} \in \text{Cl}(\mathbb{R}^{\mathbf{d+1,1}})^{\mathbf{c}}$, $X_i \in \text{AdS}^{d+1} \subseteq \mathbb{R}^{d+1,1}$.

We summarize the equivariance properties of both networks:

**Proposition 4.2.1.** After *the lifting procedure described in equation 12), **AdS-GNN** is equivariant to the global conformal group:* $\text{Conf}_{\text{g}}(\mathbb{R}^d) = \text{PO}(d + 1, 1)$*, and, **AdS-CEGNN** to the restricted conformal group:* $\text{Conf}(\mathbb{R}^d) = \text{PO}^0(d + 1, 1)$*, the identity component of* $\text{Conf}_{\text{g}}(\mathbb{R}^d)$*, see D.5.1.*

The framework above will result in features $\mathbf{h_i}$ that are manifestly invariant under the conformal group. If performing an invariant task (e.g. classification) we can now aggregate the information in a permutation-invariant manner by summing over nodes as usual at the final layer. On the other hand,

---

[10]See e.g. Section 2.5 of (Witten, 1998) for more details.

Table 1: Classification error on SuperPixel MNIST. We augment the test set with random rotations and scaling. ✗ indicates random-guess performance. Baseline results are taken from (Bekkers et al., 2024).

| Model | Error rate, % | | |
|---|---|---|---|
| | non-augmented | rotated | rotated+scaled |
| MONET | 8.89 | ✗ | ✗ |
| SplineCNN | 4.78 | ✗ | ✗ |
| GCGP | 4.2 | ✗ | ✗ |
| GAT | 3.81 | ✗ | ✗ |
| PNCNN | $1.24 \pm 0.12$ | ✗ | ✗ |
| PΘNITA | $1.17 \pm 0.11$ | 1.17 | ✗ |
| EGNN | $4.17 \pm 0.45$ | 4.17 | ✗ |
| AdS-GNN (**Ours**) | $4.09 \pm 0.27$ | 4.09 | 4.09 |

Figure 3: Test error on augmented data, SuperPixel MNIST.

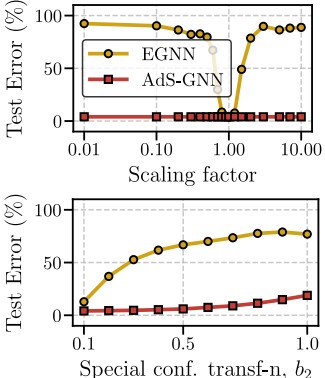

for a regression task we should specify the transformation properties of the output; e.g. if the output from the network is a conformal field $\mathbb{O}(x_i)$ living on the boundary it must transform under scale transformations with a specified $\Delta$ as in equation 3. This can be accomplished by taking the output $\mathbb{O}(x_i)$ to be

$$\mathbb{O}(x_i) = \hat{z}_i^{-\Delta} h_i^{l_{\text{final}}} \tag{17}$$

where $h_i^{l_{\text{final}}}$ is the conformally invariant output from the node associated with $x_i$ and $\hat{z}_i$ the $z$-coordinate of the embedding found in equation 12. This relation is the inverse tranformation of equation 13, and guarantees that the output satisfies equation 3.

**Computational cost** The computational cost of our approach decomposes into two components: the AdS embedding procedure (Algorithm 1) and applying the graph neural network. In the former, we employ $k$-nearest neighbors which has $\mathcal{O}(N \log N)$ complexity. An MPNN scales linearly with the number of nodes $N$ in the graph. We note that in the case when a graph is available, one potentially can use it during the lifting, thus alleviating the overhead of $k$-nearest neighbors.

## 5 EXPERIMENTAL RESULTS

We test the framework above on tasks that are loosely divided into two types: computer vision tasks and applications from physics. For most of the tasks we use the scalar AdS-GNN.

### 5.1 COMPUTER VISION TASKS

**SuperPixel MNIST** We benchmark AdS-GNN on the super-pixel MNIST dataset (Monti et al., 2017), which consists of 2D point clouds of MNIST digits segmented into 75 superpixels. Results are given in Table 1. For in-distribution data, AdS-GNN performs on par with its roto-equivariant counterparts. It does however fall slightly behind PΘNITA, which has orientational information; we feel this happens as AdS-GNN is unable to handle orientation and relies on invariant descriptors. We also study the response of a model to various augmentations (see Fig. 3); for this we compare to EGNN and find that AdS-GNN has much stronger generalization capacities. As expected, AdS-GNN is precisely scale-invariant. For special conformal transformations, there is a small breaking of symmetry arising from the uplift. We empirically measure the effect on augmented performance in Figure 3 for a special conformal transformation as in (2) parametrized by $b = (0, b_2)$ and verify that it is very small.

**Shape segmentation** We further benchmark against a shape recognition dataset, where we sample points from a selection of randomly sized and rotated shapes (either a square, a circle, or a triangle). The task is to assign a shape class to each point in the cloud. We benchmark against EGNN and a message-passing neural network MPNN which conditions messages against $x_i - x_j$ (and thus has only translational equivariance, not rotational). Here, as shown in the left panel of Figure 4 we find that AdS-GNN outperforms EGNN even on in-distribution performance. We believe that this happens as the training data contains structure at different sizes to which AdS-GNN adapts more efficiently

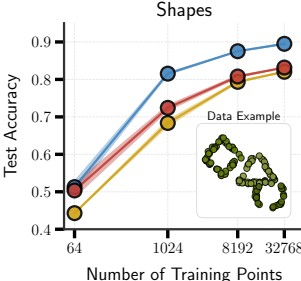
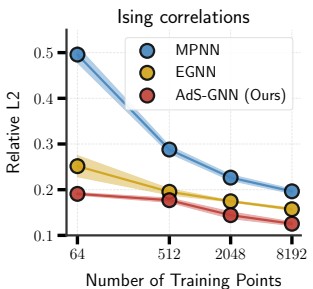
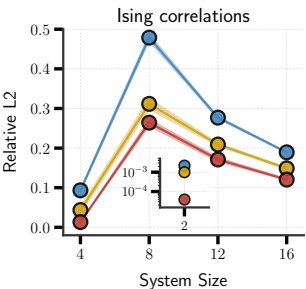

Figure 4: Performance on shape segmentation and the Ising task. *Left*, test accuracy of shape segmentation as a function of the number of training points. *Middle*, relative L2 as a function of the number of training points, with system size fixed at $N = 16$. *Right*, loss as a function of system size with 8192 training points. Inset shows $N = 2$.

than EGNN, particular when there is a small number of training points. Both of them fall behind MPNN, which we again attribute to their lack of orientational information.

A further segmentation task – PascalVOC – is discussed in the Appendix, where we find that AdS-GNN is essentially the same as EGNN.

### 5.2 PHYSICS TASKS

**2d Ising** We next consider a task from statistical physics, that of predicting $N$-point correlation functions in the 2d Ising model. As we review in more detail in Appendix C, this is a simple model of magnetism with a tunable temperature. As one increases it the model undergoes a phase transition, and at this point fluctuations of the magnetization exhibit conformal symmetry and are precisely understood in terms of a continuum model called the 2d Ising conformal field theory. This model is exactly solvable and has two non-trivial types of conformal field, the spin $\sigma(x)$ and the energy $\epsilon(x)$, which have conformal dimensions $\Delta_\sigma = \frac{1}{8}$ and $\Delta_\epsilon = 1$ respectively[11].

To benchmark AdS-GNN we consider the task of predicting the $N$-th moment of the energy and spin operators $\langle \epsilon(x_1)\epsilon(x_2)\cdots\epsilon(x_N)\rangle$ as a function of their input points. These functions are known explicitly in 2d (see equation 44) and are rather complicated: they have an erratic pattern of spikes when two points come close, on top of a more gentle modulation arising from the background of the other points.

We create a training dataset where we sample the coordinates of input points uniformly in $[-2, 2]$ for various values of $N$ and use conformal field theory results for the $N$-th moment as the training data. We simultaneously predict the spin and energy moments, using the sum of relative L2 losses. As the output is the $N$-point function of conformal fields with a non-trivial $\Delta$ we use $N$ copies of equation 17 to find our prediction $\mathrm{Pred}_a(\{x_i\})$ in each channel $a$ to be

$$\log(\mathrm{Pred}_a(\{x_i\})) = \mathrm{AdSGNN}_a(\{x_i\}) - \Delta_a \sum_{i=1}^{N} \log(\hat{z}_i) \tag{18}$$

where $a \in \{\sigma, \epsilon\}$ and the $\Delta_a$'s are trainable parameters which can be interpreted as the learned conformal dimension of the $\sigma$ and $\epsilon$ fields respectively.

We benchmark against EGNN and a baseline message-passing neural network (MPNN) whose messages are conditioned against $x_i - x_j$, and which thus only has translational equivariance. The results are shown in Figure 4 for the choice $k_{\mathrm{lift}} = 1$. We see that the performance of AdS-GNN is *superior in all regimes*. We also note various useful features. Note that for 2 points (inset on right panel of Figure 4) AdSGNN performs more than an order of magnitude better than the others. This is because the form of the two-point function is completely fixed by conformal invariance to be proportional to $|x_1 - x_2|^{-2\Delta}$, and thus AdS-GNN need only learn two numbers, whereas the others must learn to reconstruct the functional form of the power law.

---

[11]There are other slightly non-local fields that we will not discuss here.

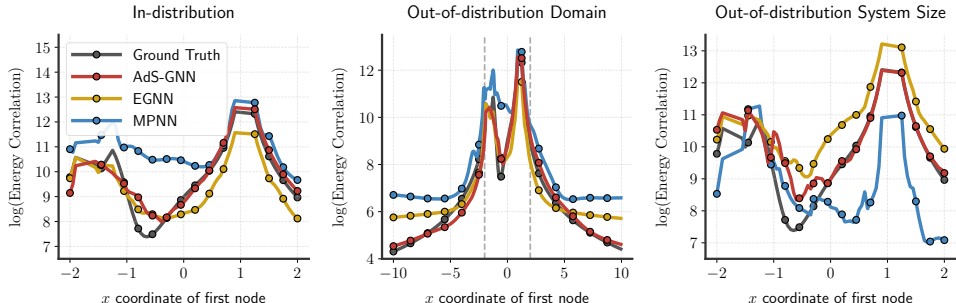

Figure 5: Visualization of the output from various models on the Ising task; here all points are fixed except for the $x$ coordinate of the first, which is varied. *Left*, models tested on in-distribution data. *Center*, testing on values of $x$ which are far out of the training range, which is shown with dashed lines. *Right*, testing a model trained on $N = 8$ on a test set with $N = 16$.

*Interpretability*: from equation 18, the learned values of the conformal dimensions $\Delta_\epsilon$ and $\Delta_\sigma$ may be read off at the end of training, as shown in Table 3. They are very close to the ground truth, showing the useful ability of this model to extract conformal dimensions from data.

*Generalization*: AdS-GNN generalizes better than EGNN in two different directions. As shown in Figure 5, it generalizes well to values of $x$ that are outside the training data, as one might have expected from scale-equivariance. AdS-GNN also generalizes well *across* $N$; as shown in an example in Figure 5, AdS-GNN trained with one value of $N$ works well when asked to predict correlation functions with a different $N$, suggesting that it has robustly understood the underlying physics. It always generalizes better than EGNN, particularly when the graph connectivity is more dense. This is discussed quantitatively in Appendix B.2.

**3d Ising** We perform a similar analysis for the 3d Ising model. This model is not exactly solvable, and there are no simple formulas for the $N$-point functions to use for training data. Nevertheless it is possible to use modern conformal bootstrap methods (reviewed in Appendix C.2) to compute a good approximation in the case $N = 4$ for the spin operator. We use this to build a training dataset and then perform the same regression as above. Our results are shown in Figure 6. Just as in the case of the 2d Ising model, AdS-GNN clearly has superior performance.

**N-body simulation** In the last experiment, we employ a neural network model for learning the dynamics of a system of 10 charged particles. The particles interact through electric fields in three spatial dimension; the inverse-square law of the electric field means that this a conformally invariant problem in 3d. The acceleration is expected to be conformal vector field with $\Delta_a = 2$ (as it is determined by an electric field, whose conformal dimension we discussed below equation 3).

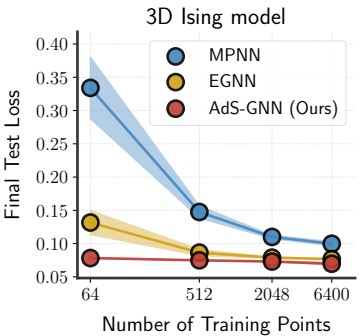

Figure 6: Relative L2 loss for the Ising 3D task. AdS-GNN also correctly recovers the conformal dimension of spin $\Delta_\sigma = 0.518$.

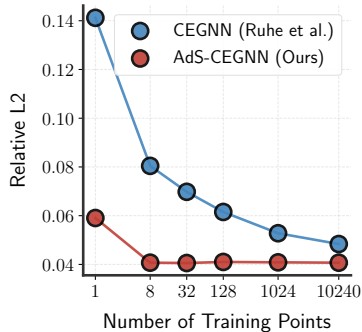

Figure 7: Relative L2 loss for the N-body task. The $L2$ error is computed across long trajectories.

We forecast acceleration value $a_t \in \mathbb{R}^{10 \times 3}$ for each body given only the position $x_t \in \mathbb{R}^{10 \times 3}$ at the time step $t$. At training, the model is learned from random slices of short ($T = 4$) training trajectories. During evaluation, we generate a set of very long trajectories (2,000 time steps) and measure the

relative $L2$ error between the ground-truth trajectories and those unrolled by the models. The task involves handling vector information, hence we use AdS-CEGNN. We benchmark against CEGNN (Ruhe et al., 2023), which is SOTA at N-body simulation.

AdS-CEGNN outperforms CEGNN, as evident from the results in Fig.7. Furthermore, it indeed correctly recovers the conformal dimension associated with acceleration $\Delta_a = 2$. Therefore, AdS-CEGNN effectively provides useful feedback when the value of the conformal dimension is known a priori: if at the end of the training $\Delta$ is correct, the model has processed the data correctly. This is in general not the case for deep learning models whose quality is based exclusively on performance.

## 6    CONCLUSION

In this paper, we introduced AdS-GNN and AdS-CEGNN - neural networks that are equivariant with respect to conformal transformations. We have demonstrated strong performance on various tasks, including computer vision and N-body simulation. We found particularly interesting an application to the Ising model, where the model exhibited impressive generalization capacities and *interpretability*, in that the conformal dimensions $\Delta_a$ – important universal quantities – could be extracted from the trained model. One might ask what other such universal information exists: a general conformally invariant field theory in physics is defined in terms of these dimensions $\Delta_a$ and a set of 3-point coefficients $c_{abc}$ which turn out to determine higher-order moments of the conformal fields [12]. It is interesting to ask whether this information could be be usefully extracted from the trained network. We anticipate further applications to critical phenomena and long-range interactions in physical systems.

**Future Work**    Besides physics-related tasks, we see potential applications of AdS-GNN and AdS-CEGNN to computed vision and robotics tasks due to the inherent consistency of our models to local scale variations. That is, proposed models maintain consistent predictions when objects appear at varying scales within scenes, and do not require extensive multi-scale training data to achieve robustness.

### ACKNOWLEDGMENTS

MZ was supported by Microsoft Research AI4Science. This work was supported by a grant from the Simons Foundation (PD-Pivot Fellow-00004147, NI and EB). NI is supported in part by the STFC under grant number ST/T000708/1. This publication is part of the project SIGN with file number VI.Vidi.233.220 of the research programme Vidi which is (partly) financed by the Dutch Research Council (NWO) under the grant https://doi.org/10.61686/PKQGZ71565

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

# A $\mathrm{AdS}^{d+1}$ DETAILS

We provide a few further details about aspects of $\mathrm{AdS}^{d+1}$:

## A.1 THE GROUP ACTION ON AdS

Recall that $\mathrm{AdS}^{d+1}$ can be understood as the hyperboloid defined in equation 5. The action of $[\Lambda] \in \mathrm{O}(d+1,1)/\{\pm I\}$ then works concretely as follows

$$[\Lambda].Y := \mathrm{sign}((\Lambda Y)^0) \cdot (\Lambda Y) \tag{19}$$

where $\Lambda Y$ is just vector matrix multiplication and where the scalar multiplication with $\mathrm{sign}((\Lambda Y)^0)$ corrects the sign of the 0-th entry (the component of $(-1)$-signature). Together we get:

$$((([\Lambda].Y)^0)^2 - \sum_{i=1}^{d+1}(([\Lambda].Y)^i)^2 = 1, \qquad ([\Lambda].Y)^0 > 0, \tag{20}$$

and thus: $[\Lambda].Y \in \mathrm{AdS}^{d+1}$. This gives us a well-defined group action of $\mathrm{O}(d+1,1)/\{\pm I\}$ on $\mathrm{AdS}^{d+1}$, and, in particular, a well-defined group action of $\mathrm{Conf}(\mathbb{R}^d)$ on $\mathrm{AdS}^{d+1}$. Note, however, that this action does not act linearly (since $-I$ acts trivially). This shows that we have now two different group actions: i) the linear action, given just by matrix multiplication, and ii) the sign-corrected action, given by Equation 19. It is a subtle, but important point to notice that for every $\Lambda \in \mathrm{O}^0(d+1,1)$, the connected component of the identity $I$, we have $\mathrm{sign}((\Lambda Y)^0) = 1$ whenever $Y^0 > 0$. So for those elements $\Lambda \in \mathrm{O}^0(d+1,1)$ these two actions agree. Note that $-I \notin \mathrm{O}^0(d+1,1)$, see D.1.5. Since we can canonically identify the (restricted) conformal group with the above:

$$\mathrm{Conf}(\mathbb{R}^d) = \mathrm{PO}^0(d+1,1) \cong \mathrm{O}^0(d+1,1), \tag{21}$$

we get a well-defined *linear* action of $\mathrm{Conf}(\mathbb{R}^d)$ on $\mathrm{AdS}^{d+1}$, meaning that the inclusion map:

$$j : \mathrm{AdS}^{d+1} \subseteq \mathbb{R}^{d+1,1} \tag{22}$$

is equivariant w.r.t. $\mathrm{Conf}(\mathbb{R}^d)$, considered as the subgroup $\mathrm{Conf}(\mathbb{R}^d) \cong \mathrm{O}^0(d+1,1) \subseteq \mathrm{O}(d+1,1)$. This implies that any $\mathrm{O}(d+1,1)$-equivariant map (like CGENNs) $\psi : \mathbb{R}^{d+1,1} \to \mathcal{H}$ induces a $\mathrm{Conf}(\mathbb{R}^d)$-equivariant map:

$$\psi \circ j : \mathrm{AdS}^{d+1} \to \mathbb{R}^{d+1,1} \to \mathcal{H}. \tag{23}$$

This insight will be used to construct conformally equivariant graph neural networks (with multivector features).

## A.2 THE CENTER OF MASS OF A SET OF POINTS ON ADS

We will require an expression for the "center of mass" $C(\{X_i\})$ of a set of points on AdS. This problem was solved in (Galperin, 1993); the basic idea is to view the hyperboloid as a submanifold of $\mathbb{R}^{d+1,1}$ as above, use additivity properties there to find a vector, and then find the intersection of the ray in the direction of that vector with the hyperboloid.

In practice, this is quite simple to implement. Denote the center of mass by $\bar{Y}^A$, and the set of $N$ points for which we want the centroid by $(x_i^a, z_i)$. We would like to find the analogous coordinates for the centroid $(\bar{x}^a, \bar{z})$.

We have that

$$\bar{Y}^0 \equiv \frac{\bar{z}}{2}\left(1 + \frac{1}{\bar{z}^2}(1 + \sum_a \bar{x}^a \bar{x}^a)\right) = \frac{1}{\mathcal{N}N}\sum_i \frac{z_i}{2}\left(1 + \frac{1}{z_i^2}(1 + \sum_a x_i^a x_i^a)\right) \tag{24}$$

$$\bar{Y}^a \equiv \frac{\bar{x}^a}{\bar{z}} = \frac{1}{\mathcal{N}N}\sum_i \frac{x_i^a}{z_i} \tag{25}$$

$$\bar{Y}^{d+1} \equiv \frac{\bar{z}}{2}\left(1 - \frac{1}{\bar{z}^2}(1 - \sum_a \bar{x}^a \bar{x}^a)\right) = \frac{1}{\mathcal{N}N}\sum_i \frac{z_i}{2}\left(1 - \frac{1}{z_i^2}(1 - \sum_a x_i^a x_i^a)\right) \tag{26}$$

The first equality is the definition of the embedding, the second is the definition of the centroid from Galperin. Here $\mathcal{N}$ is a normalization constant which is picked to guarantee that

$$(\bar{Y}^0)^2 - \sum_a (\bar{Y}^a)^2 - (\bar{Y}^{d+1})^2 = 1 \tag{27}$$

So to find the centroid, the easiest thing to do is to compute the sums on the right hand side of the second equality, which thus determines the vector $\bar{Y}^A$ up to an overall scale $\mathcal{N}$; then we enforce the norm constraint above which lets us find $\mathcal{N}$ and thus fixes the vector $\bar{Y}^A$ completely. We then express the answer in useful coordinates by solving for $(\bar{z}, \bar{x}^a)$ through

$$\bar{z} = \frac{1}{\bar{Y}^0 - \bar{Y}^{d+1}}, \qquad \bar{x}^a = \frac{\bar{Y}^a}{\bar{Y}^0 - \bar{Y}^{d+1}}. \tag{28}$$

To get some intuition for the procedure, we study it in the case of two points $X_1 = (x_1^a, \epsilon)$ and $X_2 = (x_2^a, \epsilon)$ starting at the same value of the $z$ coordinate. We find

$$C(X_1, X_2) = \left(\frac{1}{2}(x_1^a + x_2^a), \frac{1}{2}\sqrt{|x_1 - x_2|^2 + 4\epsilon^2}\right) \tag{29}$$

i.e. we simply take the average of the spatial coordinates and move inwards in $z$ by an amount which depends on the separation between the two points in the spatial direction. In this case the center of mass is actually the midpoint of the geodesic that connects the two points.

### A.3 EQUIVARIANCE OF LIFTING PROCEDURE

Here we prove Proposition 4.1.1, i.e. that the lifting procedure is equivariant under translations, rotations and dilatations but not special conformal transformations. We denote a general element of the conformal group by $G$. $G$ acts on points $x_i^a \in \mathbb{R}^d$ as in equation 2 and on points in $X_i^\mu \in \mathrm{AdS}^{d+1}$ as in equation 8.

The lifting begins by embedding a set of points $\{x_i\} \in \mathbb{R}^d$ into $\mathrm{AdS}^d$ using a small regulator value $z = z_0$ as in equation 11:

$$X_{i,0}^\mu(x_i^a) := (x_i^a, z = z_0) \tag{30}$$

We then compute the AdS center of mass $C(\{X_{i,0}\})$ of the set of points using the algorithm above. The question is whether this approach is equivariant in the bulk when the conformal transformation acts *only* on the boundary coordinates $x^a$, i.e. is it true that

$$C(\{X_{i,0}(Gx_i^a)\}) \stackrel{?}{=} GC(\{X_{i,0}(x_i^a)\}) \tag{31}$$

(Note that if the conformal transformation also acted on $z_0$ then this would certainly be equivariant; the question at hand is how much the choice of fixed regulator $z_0$ spoils the equivariance).

It is immediately clear that the procedure is equivariant under translations and rotations, as in those cases we see from equation 8 that the $z$ coordinate is left invariant under the transformation. Through direct computation using equation 2 and equation 8, we can further explicitly verify that the equality in equation 31 holds for dilatations but does *not* hold for special conformal transformations.

This may seem somewhat surprising, and we try to understand it more abstractly by studying the geometry of the embedded points $X_{i,0}$ in the $Y^A$ coordinates defined in equation 6:

$$Y_i^0 := \frac{z_0^2 + \|x_i\|^2 + 1}{2z_0}, \qquad Y_i^a := \frac{x_i^a}{z_0}, \qquad Y_i^{d+1} := \frac{z_0^2 + \|x_i\|^2 - 1}{2z_0} \tag{32}$$

As we take $z_0 \to 0$ while holding $x_i$ fixed, we can extract the dependence on $z_0$ by writing these points as $Y_i^A = \frac{1}{z_0} Y_{i,\text{null}}^A$. Explicitly we then have

$$Y_{i,\text{null}}^0 := \frac{\|x_i\|^2 + 1}{2}, \qquad Y_{i,\text{null}}^a := x_i^a, \qquad Y_{i,\text{null}}^{d+1} := \frac{\|x_i\|^2 - 1}{2} \qquad (33)$$

Here the points $Y_{i,\text{null}}^A$ lie on the null cone defined by $\|Y\| = 0$, where the norm is taken with respect to the $\mathbb{R}^{d+1,1}$ metric defined in equation 4. The points on this cone no longer depend explicitly on $z_0$, and furthermore the cone has a natural $O(d+1,1)$ action on it given by regular matrix multiplication on $Y^A$ as in equation 19.

Naively, one would now expect that one could just ignore the overall normalizing factor and sum up the points in the cone when computing the sum in equation 24. As $Y^0 > 0$ for all points this would lead to a point in the *interior* of the null cone (i.e. in $\text{AdS}^{d+1}$ itself) and the procedure now appears to make no reference to $z_0$ and is manifestly equivariant under all of $O(d+1,1)$.

Interestingly, this is not the case. It turns out that under $x^a \to Gx^a$ the points on the cone do not transform with the usual action of $O(d+1,1)$; instead they acquire an overall scalar rescaling factor $\sigma$ which depends on the parameters of $G$ and the point in question, i.e.

$$GY_{i,\text{null}}(x_i^a) = \sigma(G, x^a) Y_{i,\text{null}}(Gx_i^a) \qquad (34)$$

The scalar factor preserves the cone $\|Y\| = 0$, but it means that the sum in equation 24 no longer transforms equivariantly.

The existence of this factor can already be seen in the simplest case $d = 1$, where the embedding simply becomes

$$Y_i^0 := \frac{z_0^2 + x^2 + 1}{2z_0}, \qquad Y_i^1 := \frac{x}{z_0}, \qquad Y_i^2 := \frac{z_0^2 + x^2 - 1}{2z_0} \qquad (35)$$

We can now compute the prefactor $\sigma$ for various transformations by computing both sides of equation 34. Under a dilatation $x \to \lambda x$ the overall prefactor $\sigma(\lambda, x)$ is simply a constant $\lambda^{-1}$ which can indeed be ignored when computing the center of mass; thus the lifting procedure is equivariant under dilatations. However under a special conformal transformation $x \to \frac{x}{bx+1}$ the prefactor is instead $\sigma(b, x) = (1 + bx)^2$ and thus depends on the point in question. This spoils the equivariance under special conformal transformations.

Our discussion here is somewhat formal but the conclusion is expected on physical grounds, as in general it is difficult to preserve full conformal invariance in the presence of a regulator such as $z_0$.

### A.4 VECTOR BULK-TO-BOUNDARY MAPPING

Given an $O(d+1,1)$ vector feature $v_A^{\text{bulk}}$ expressed in the $Y^A$ coordinates in the bulk we may need to map it back to the boundary to interpret it as a neural network output. This is first done by using the Jacobian associated with equation 6 to map it to an orthonormal basis in the $(x^a, z)$ coordinates:

$$v_{\hat{\mu}}^{\text{bulk}} = e_{\hat{\mu}}^{\mu} \frac{\partial Y^A}{\partial X^\mu} v_A^{\text{bulk}} \qquad (36)$$

where we have used the Einstein summation convention, where $X^\mu = (x^a, z)$, we have denoted and where $e_{\hat{\mu}}^{\mu}$ is the inverse vielbein which satisfies:

$$e_{\hat{\mu}}^{\mu} e_{\hat{\nu}}^{\nu} g_{\mu\nu} = \delta_{\hat{\mu}\hat{\nu}} \qquad (37)$$

where $g_{\mu\nu}$ is the $\text{AdS}^{d+1}$ metric. In the coordinates used in equation 7 this vielbein is simply $e_{\hat{\mu}}^{\mu} = z\delta_{\hat{\mu}}^{\mu}$. Finally the analogue of equation 17 is

$$v_{\hat{\mu}}^{\text{output}} = z^{-\Delta} v_{\hat{\mu}}^{\text{bulk}} \qquad (38)$$

Working out the Jacobian explicitly we find that the spatial components of the vector are

$$v_{\hat{a}}^{\text{output}} = z^{-\Delta} \left( v_a^{\text{bulk}} + x_a(v_{d+1}^{\text{bulk}} + v_0^{\text{bulk}}) \right) \qquad (39)$$

where we stress that the components on the right-hand side are of the vector $v_A^{\text{bulk}}$ expressed in the $Y^A$ coordinates, as is natural for the Clifford algebra construction.

## B    FURTHER EXPERIMENTAL DETAILS

### B.1    PASCALVOC-SP

In a further experiment, we also compare AdS-GNN to EGNN on the LRGB data (Dwivedi et al., 2022), see Table 2. This is a pixel segmentation task; thus the output data at each node is conformally *in*variant and we take $\Delta = 0$ in equation 17, i.e. we read out the output from $\mathbf{h}_i$ directly. The difference in performance between EGNN and AdS-GNN is statistically insignificant, which indicates that in this task conformal equivariance does not constrain the model significantly and still allows for high expressivity.

Table 2: Classification error on PascalVOC-SP.

| Model | EGNN | AdS-GNN |
|---|---|---|
| Test F1 $\uparrow$ | $27.80 \pm 0.74$ | $28.07 \pm 0.57$ |

### B.2    2D ISING GENERALIZABILITY

Here we discuss further the generalization properties of the 2d Ising task discussed in the main text. In particular, we provide information on how a a model trained on a given number of nodes $N_{\text{train}}$ performs when evaluated on a different number of nodes $N_{\text{test}}$. The connectivity on the graph on AdS is controlled by a parameter $k_{\text{con}}$, the number of nearest neighbours which are connected to each node.

| System size | $\Delta_\epsilon$ (energy) | $\Delta_\sigma$ (spin) |
|---|---|---|
| 2 | $1.0000 \pm 0.0000$ | $0.1250 \pm 0.000$ |
| 4 | $0.9998 \pm 0.0000$ | $0.1250 \pm 0.000$ |
| 8 | $0.9924 \pm 0.0010$ | $0.1248 \pm 0.000$ |
| 12 | $0.9894 \pm 0.0032$ | $0.1247 \pm 0.000$ |
| 16 | $0.9893 \pm 0.0007$ | $0.1247 \pm 0.000$ |

Table 3: Learned value of $\Delta$'s from AdS-GNN. Ground truth values are $\Delta_\epsilon = 1$ and $\Delta_\sigma = 0.125$. Statistical uncertainties are standard deviations from 5 runs; for $\Delta_\sigma$ they are $O(10^{-5})$ and are not quoted.

For most of our results in the Ising section, we pick $k_{\text{con}} = N$ so that the graph is fully connected. This results in the best in-distribution performance. In that case note from Figures 8a and 8b that AdS-GNN generalizes dramatically better than EGNN. EGNN's performance is particularly bad when taking a model trained on a larger system and evaluating it on a smaller one.

However if we reduce $k_{\text{con}}$ – e.g. to 2 – then this greatly increases the generalization ability of both models. In particular, the gap between EGNN and AdS-GNN (as seen in Figures 8c and 8d) is much narrower, though still present. We see however that this choice hurts in-distribution performance; AdS-GNN is able to generalize reasonably well even in the fully connected case.

### B.3    IMPLEMENTATION DETAILS

In every experiment, we use the AdamW optimizer (Loshchilov & Hutter, 2019) with a learning rate $10^{-3}$. Every model is trained on a single Nvidia RTX6000 GPU. All models are implemented in JAX. All experiments are run 5 times with different seeds. Models are trained until convergence with early stopping. The number of layers is fixed to 4, with hidden dimension 32.

**SuperPixel-MNIST**    The task is to predict a digit given a point cloud representation. We compare against MONET (Monti et al., 2017), SplineCNN (Fey et al., 2018), GCCP (Walker & Glocker, 2019), GAT (Velickovic et al., 2018), PNCNN (Finzi et al., 2021) and PΘNITA (Bekkers et al., 2024). Every model is trained with batch size 128, baseline results are taken from (Bekkers et al., 2024). $k_{\text{con}}$ is set to 16, $k_{\text{lift}}$ to 5. Training time is approximately 10 minutes.

**Pascal-VOC**    The task is to predict a semantic segmentation label for each superpixel node (total of 21 classes). Each graph is embedded in 2D Euclidean space, each node is associated with 12 scalar features. We used the batch size of 96. $k_{\text{con}} = 16$, $k_{\text{lift}} = 5$. Training time is approximately 30 minutes.

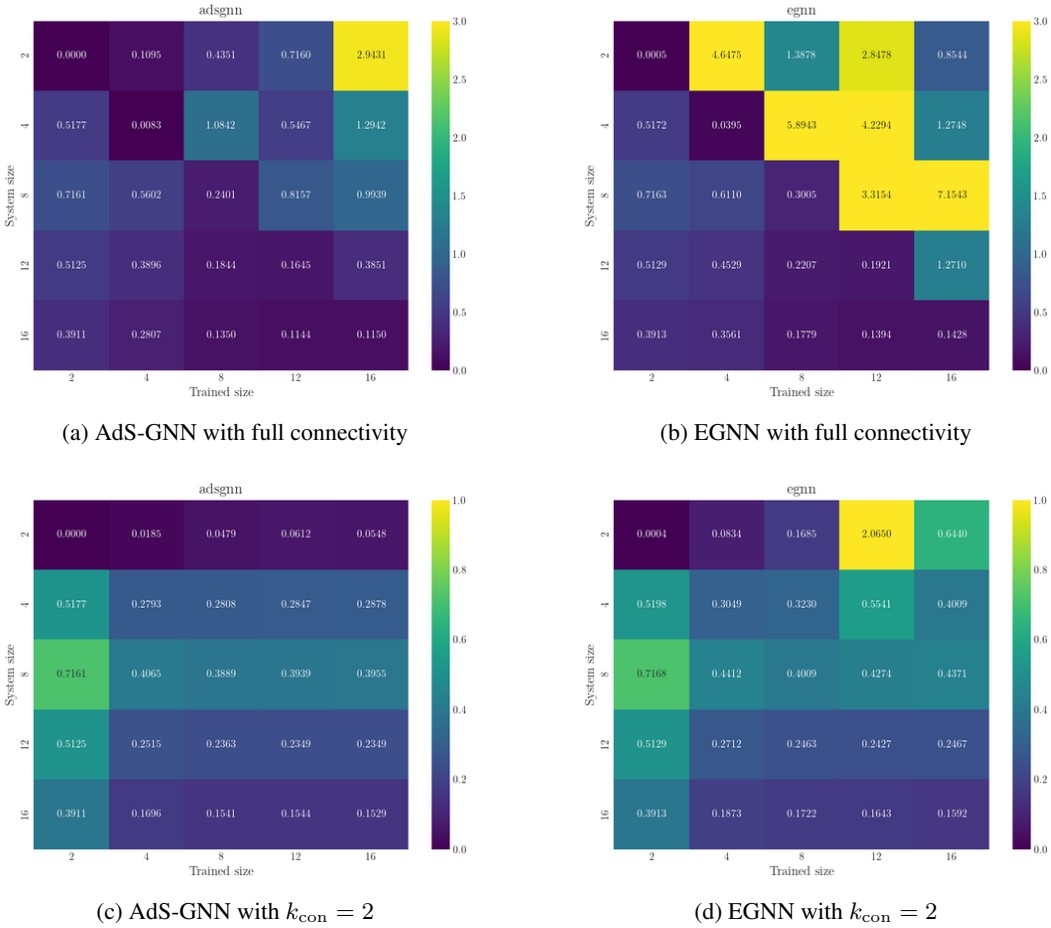

Figure 8: Generalization across system size; each square shows the relative L2 loss of a system trained on a system of size $N_{\text{train}}$ ($x$-axis) when tested on a system of size $N_{\text{test}}$ ($y$-axis) for different level of graph connectivity.

**Shapes** Given a point cloud, we predict for each point to each shape it belongs (circle, square, triangle, intersection). We train by minimizing cross-entropy loss. $k_{\mathrm{con}}$ is set to 16, $k_{\mathrm{lift}} = 16$. The number of testing and validation points is set to 512, while the number of training points varies from 64 to 8192. Training time is approximately 5 minutes.

**Ising** Given a set of points, we predict two scalar values, one for energy correlation, one for spin correlation. We use relative L2 error as our training objective. $k_{\mathrm{con}}$ is equal to the number of points (i.e. fully connected system), $k_{\mathrm{lift}} = 1$. The number of testing and validation points is set to 512, while the number of training points varies from 64 to 32768. Training time is approximately 3 minutes.

## C    ISING MODEL REVIEW

Here, for completeness, we provide an elementary review of the Ising model – which can be imagined as the simplest example of a model of magnetism – and the physics at its critical point. This material is standard; see e.g. (Kardar, 2007; Di Francesco et al., 1997) for textbook treatments. We begin by discussing the model in 2d.

### C.1    2D ISING MODEL

The model is defined in terms of binary variables called *spins* $\sigma_i = \pm 1$ sitting on the sites $i$ of a square lattice with $L$ sites on a side. The model is defined in terms of an energy function:

$$E[\sigma] = -\sum_{\langle ij \rangle} \sigma_i \sigma_j \tag{40}$$

where the notation $\langle ij \rangle$ means that one sums over nearest-neighbour links connecting two adjacent sites $i, j$. We can see that the energy is minimized when spins on two adjacent sites have the same value, i.e. "spins want to align". The energy is also invariant under a $\mathbb{Z}_2$ symmetry which acts by flipping all of the spins, $\sigma_i \to -\sigma_i$.

This energy defines a statistical physics model in which the probability of obtaining a given spin configuration $\{\sigma\}$ is given by

$$p_\beta[\sigma] = \frac{1}{Z} \exp(-\beta E[\sigma]) \tag{41}$$

where $\beta$ is the inverse temperature and $Z$ the usual normalizing constant. This model has two *phases*, which we now describe.

Consider first taking $\beta$ very large; in that case any increase in energy will be heavily penalized, and the most likely configurations will be those that minimize the energy, i.e. where all spins have the same value, so either $\sigma_i = +1$ or $\sigma_i = -1$ for all $i$. This is called a phase with *spontaneous symmetry breaking*, as a choice of either of these configurations breaks the $\mathbb{Z}_2$ symmetry. It is also called the *ordered* or *ferromagnetic* phase.

Now consider taking $\beta$ very small; in that case the system is very disordered, and all spins fluctuate strongly and randomly, and there is no sense in which a symmetry is spontaneously broken. This is called the *symmetry unbroken* or *disordered* phase or the *paramagnet.*.

In the $L \to \infty$ limit there is a sharp distinction between the two phases. A quantitative way to understand it is to imagine taking the system size to infinity while computing e.g.

$$\langle \sigma \rangle = \lim_{L \to \infty} \left( \frac{1}{L^2} \mathbb{E}_{\sigma \sim p_\beta} \sum_i \sigma_i \right) \tag{42}$$

i.e. the expectation value of the spatial average of all the spins. This is called the order parameter. This is nonzero in the ordered phase (where all the spins are aligned, resulting in a net contribution to the expectation value) and zero in the disordered phase (where all the spins fluctuate strongly, resulting in a cancellation across sites). In the infinite-$L$ limit there is a non-analyticity in the function $\langle \sigma(\beta) \rangle$ at a critical value of $\beta = \beta_c$ at the phase transition point. For the 2d Ising model the location of this point is known to be at $\beta_c = \frac{1}{2} \log(1 + \sqrt{2}) \approx 0.441$.

A great deal is known about this critical point. Here fluctuations of the spins take place over all scales, and do not decay exponentially with distance as one might normally expect. It is possible to capture the long-distance statistics of these fluctuations in a continuum limit where we formally take the lattice spacing to zero. The resulting structure is called the *2d Ising conformal field theory* (CFT), and is an example of a quantum field theory that exhibits conformal symmetry. In particular, one can define two operators in this continuum theory: the spin operator $\sigma(x)$ (which is the continuum limit of the spin operator $\sigma_i$ defined on discrete lattice sites above) and the energy operator $\epsilon(x)$ (which can be thought of as a product of spins at adjacent sites). The 2d Ising CFT is completely solved and thus one can compute any arbitrary moments of any of these operators in closed form.

To get some intuition, the correlation function of two spins behaves as:

$$\langle \sigma(x)\sigma(y) \rangle = |x-y|^{-2\Delta_\sigma} \tag{43}$$

where $\Delta_\sigma = \frac{1}{8}$. The power-law functional form is completely fixed by scale-invariance, and the only input from the theory here is the value of the conformal dimension $\Delta_\sigma$. A similar relation holds for the energy operator with $\Delta_\epsilon = 1$.

In this work we build a neural network to predict the $N$-point correlation functions of the spin and energy operators as a function of the positions of the operator insertions. To do this we use as ground-truth training data the following closed-form formulas from the theory of the 2d Ising CFT (Di Francesco et al., 1997):

$$\langle \epsilon(z_1)\epsilon(z_2)\cdots\epsilon(z_N) \rangle = \left| \mathrm{Pf} \left[ \frac{1}{z_i - z_j} \right]_{1 < i,j < N} \right|^2 \tag{44}$$

$$\langle \sigma(z_1)\sigma(z_2)\cdots\sigma(z_N) \rangle^2 = \frac{1}{2^{2N}} \sum_{\epsilon_i = \pm 1, \sum_{\epsilon_i} = 0} \prod_{i<j} |z_i - z_j|^{\frac{\epsilon_i \epsilon_j}{2}} \tag{45}$$

where Pf denotes the matrix Pfaffian. Despite their compact presentation, these are rather complicated functions; e.g. combinatorially the Pfaffian can be viewed as a sum over all possible perfect matchings of $N$ points. The number of these matchings grows factorially in $N$, and each of them contributes a (different) product of 2-body interactions. There are more efficient ways than this to actually *compute* the Pfaffian, but our neural networks will be unable to realize that structure.

## C.2 3D ISING MODEL

The lattice construction equation 40 may easily be generalized to 3d. Unlike the 2d Ising model, the 3d Ising model has not been explicitly solved, and there are no simple exact formulas such as equation 45 for moments of the spin operator at the critical point. Nevertheless, a great deal is known about it, where the most precise information comes from the conformal bootstrap (El-Showk et al., 2012), i.e. a method which exploits the self-consistency of conformal field theory to solve the system. For reviews see (Simmons-Duffin, 2017; Poland et al., 2019).

Using these ideas, the 4-point function of the spins is given by

$$\langle \sigma(x_1)\sigma(x_2)\sigma(x_3)\sigma(x_4) \rangle = \frac{g(u,v)}{|x_1 - x_2|^{2\Delta_\sigma}|x_3 - x_4|^{2\Delta_\sigma}} \tag{46}$$

where the scaling dimension of the spin operator is $\Delta_\sigma = 0.5181489(10)$. $u, v$ are conformally invariant functions of the four insertion points called *cross-ratios*, and are

$$u \equiv \frac{x_{12}^2 x_{34}^2}{x_{13}^2 x_{24}^2} \qquad v \equiv \frac{x_{23}^2 x_{14}^2}{x_{13}^2 x_{24}^2} \qquad x_{ij} \equiv x_i - x_j \tag{47}$$

The non-trivial information in this 4-point function is stored in the dependence on the cross-ratios $g(u,v)$. We compute this following (Rychkov et al., 2017), which we review briefly below.

This function $g(u,v)$ can be decomposed in the following fashion:

$$g(u,v) = \sum_{\mathbb{O} \in \sigma \times \sigma} C_{\sigma\sigma\mathbb{O}}^2 g_{\Delta_\mathbb{O}, l_\mathbb{O}}(u,v) \tag{48}$$

where the sum over $\mathbb{O}$ runs over the conformal primaries of the theory, $\Delta_\mathbb{O}$ and $l_\mathbb{O}$ are the conformal dimension and spin of the operators respectively, and $g_{\Delta_\mathbb{O}, l_\mathbb{O}}(u,v)$ is a theory-independent function

called a *conformal block*. which These functions are the same for all theories, and thus the information characterizing this function is encoded in the set $\{\Delta_{\mathbb{O}}, l_{\mathbb{O}}, C_{\sigma\sigma\mathbb{O}}\}$, where $C_{\sigma\sigma\mathbb{O}}$ is a set of theory-specific numbers called *operator product expansion coefficients*. For the 3d Ising model this set of numbers is not known analytically, but they have been computed for the operators with lowest dimension to extremely high accuracy.

Following the method in (Rychkov et al., 2017), and using the lowest five operators and their associated $\{\Delta_{\mathbb{O}}, l_{\mathbb{O}}, C_{\sigma\sigma\mathbb{O}}\}$ from (Komargodski & Simmons-Duffin, 2017), we compute $g(u, v)$ using the Mathematica code of (Costa et al., 2016) to find the conformal blocks. We then create a dataset of 4-point functions precisely as for the 2d Ising model above. This is not an exact solution to the 3d Ising model, but the error from truncating the operator sum has been conservatively estimated to be one part in $10^{-2}$ in (Rychkov et al., 2017). (We also note that for benchmarking our results presumably *any* choice of $g(u, v)$ would suffice, but we think it is more physically reasonable to use the function that is relevant to a known physical problem).

Such methods would be much harder to implement for higher-point functions, and thus our test dataset for the 3d Ising model is restricted to the 4-point function.

## D    THE CONFORMAL GROUP OF THE PSEUDO-EUCLIDEAN SPACE

In this section we will provide a self-contained introduction to conformal geometry of the (pseudo-) Euclidean space in arbitrary signature and its conformal group. We provide definitions, examples and main theorems.

We follow the references of Schottenloher (2008) and McKay (2023).

### D.1    CONFORMAL TRANSFORMATIONS OF THE PSEUDO-EUCLIDEAN SPACE

Roughly speaking, a conformal map is a map that preserves local angles. However, this definition is rather tedious to write down and work with. A technically more convenient definition is introduced below in Definition D.1.2. To motivate this definition, we first show in Lemma D.1.1 that in the (linear) Euclidean case these two definitions are equivalent:

**Lemma D.1.1.** *Let $A \in \mathbb{R}^{d \times d}$ be a real invertible $(d \times d)$-matrix. Then the following statements are equivalent:*

1. *$A$ is angle preserving, i.e. for all $v_1, v_2 \in \mathbb{R}^d \setminus \{0\}$ we have:*

$$\frac{\langle Av_1, Av_2 \rangle}{\|Av_1\| \cdot \|Av_2\|} = \frac{\langle v_1, v_2 \rangle}{\|v_1\| \cdot \|v_2\|}. \tag{49}$$

2. *$A$ is a conformal matrix, i.e. $A = c\Lambda$ with a scalar $c > 0$ and an orthogonal matrix $\Lambda \in \mathrm{O}(d)$.*

*Proof.* "$\Longleftarrow$": Let $A = c\Lambda$ with $c > 0$ and $\Lambda \in \mathrm{O}(d)$. Then we get for all $v_1, v_2 \in \mathbb{R}^d$:

$$\langle Av_1, Av_2 \rangle = v_1^\top A^\top A v_2 \tag{50}$$
$$= c^2 \cdot v_1^\top \underbrace{\Lambda^\top \Lambda}_{=I} v_2 \tag{51}$$
$$= c^2 \cdot v_1^\top v_2 \tag{52}$$
$$= c^2 \cdot \langle v_1, v_2 \rangle. \tag{53}$$

This also implies:

$$\|Av_1\| = c \cdot \|v_1\|, \qquad\qquad \|Av_2\| = c \cdot \|v_2\|. \tag{54}$$

Together this implies the claim:

$$\frac{\langle Av_1, Av_2 \rangle}{\|Av_1\| \cdot \|Av_2\|} = \frac{\langle v_1, v_2 \rangle}{\|v_1\| \cdot \|v_2\|}. \tag{55}$$

"$\Longrightarrow$": Assume that $A$ is invertible and preserves angles. Then $C := A^\top A$ is symmetric and positive definite. By the spectral theorem we can diagonalize $C$, i.e. there exists an orthonormal basis $e_1, \ldots, e_d \in \mathbb{R}^d$, $\langle e_i, e_j \rangle = \delta_{i,j}$ for all $i, j \in [d]$, and positive scalars $\lambda_1, \ldots, \lambda_d > 0$ such that for all $i \in [d]$:

$$Ce_i = \lambda_i \cdot e_i. \tag{56}$$

We now claim that: $\lambda_1 = \cdots = \lambda_d$. By way of contradiction assume that there exists $\ell \in [d]$ such that $\lambda_1 \neq \lambda_\ell$. Then put: $v_1 := e_1$ and $v_2 := e_1 + e_\ell$. This gives:

$$\|v_1\|^2 = 1, \qquad\qquad \|v_2\|^2 = \|e_1\|^2 + \|e_\ell\|^2 = 2, \tag{57}$$

$$\|Av_1\|^2 = \lambda_1, \qquad\qquad A^\top Av_2 = \lambda_1 \cdot e_1 + \lambda_\ell \cdot e_\ell, \tag{58}$$

$$\langle Av_1, Av_2 \rangle = \lambda_1, \qquad\qquad \|Av_2\|^2 = \lambda_1 + \lambda_2, \tag{59}$$

$$\langle v_1, v_2 \rangle = 1. \tag{60}$$

This implies:

$$\frac{\lambda_1}{\sqrt{\lambda_1} \cdot \sqrt{\lambda_1 + \lambda_\ell}} = \frac{\langle Av_1, Av_2 \rangle}{\|Av_1\| \cdot \|Av_2\|} = \frac{\langle v_1, v_2 \rangle}{\|v_1\| \cdot \|v_2\|} = \frac{1}{1 \cdot \sqrt{2}}. \tag{61}$$

Squaring and solving for $\lambda_\ell$ shows:

$$\lambda_\ell = \lambda_1, \tag{62}$$

which contradicts the assumption. So, we indeed have that:

$$c^2 := \lambda_1 = \cdots = \lambda_d > 0. \tag{63}$$

This shows that: $C = c^2 I$. Putting $\Lambda := \frac{1}{c}A$ we get:

$$\Lambda^\top \Lambda = \frac{1}{c^2}C = I, \tag{64}$$

which shows that $\Lambda \in \mathrm{O}(d)$ and thus $A = c\Lambda$ with $c > 0$ and $\Lambda \in \mathrm{O}(d)$. $\qquad\square$

**Definition D.1.2** (Conformal maps/conformal transformations). *Let $(M, \eta^M)$ and $(N, \eta^N)$ be two pseudo-Riemannian manifolds. A* conformal map $f : M \to N$ *is defined to be a smooth map such that there exists a smooth map $\omega : M \to \mathbb{R}_{>0}$ such that for all $x \in M$ and $v_1, v_2 \in \mathrm{T}_x M$ we have:*

$$\eta^N_{f(x)}(df_x(v_1), df_x(v_2)) = \omega(x)^2 \cdot \eta^M_x(v_1, v_2), \tag{65}$$

*where $df_x : \mathrm{T}_x M \to \mathrm{T}_{f(x)} N$ is the differential of $f$ at $x \in M$. The above map $\omega$ is called the* conformal factor *of $f$.*

*In case the conformal factor $\omega$ of $f$ equals the constant one, $\omega(x) = 1$, then we call $f$ an* isometric map *or isometric transformation.*

**Definition D.1.3** (Conformal diffeomorphisms and isometries). *Let $(M, \eta^M)$ be a pseudo-Riemannian manifold. A* conformal diffeomorphism $f : M \to M$ *is a conformal smooth map that has a conformal smooth inverse $f^{-1} : M \to M$:*

$$f \circ f^{-1} = f^{-1} \circ f = \mathrm{id}_M. \tag{66}$$

*If, in addition, its conformal factor $\omega$ of $f$ is the contant 1, then we call $f$ an* isometry of $(M, \eta^M)$. *The* group of conformal diffeomorphisms *of $(M, \eta^M)$ is denoted as:*

$$\mathrm{ConfDiff}(M, \eta^M) := \{f : M \to M \text{ conformal map with conformal inverse}\}. \tag{67}$$

*The* isometry group *of $(M, \eta^M)$ is denoted as:*

$$\mathrm{Isom}(M, \eta^M) := \{f : M \to M \text{ isometric map with isometric inverse}\}. \tag{68}$$

**Notation D.1.4.** *In the following we will denote by:*

1. *$(\mathbb{R}^{p,q}, \eta^{p,q})$ be the* standard pseudo-Euclidean space *of signature $(p, q)$:*

$$\eta^{p,q}(v_1, v_2) := v_1^\top \Delta^{p,q} v_2, \qquad \Delta^{p,q} := \mathrm{diag}(\underbrace{+1, \ldots, +1}_{\times p}, \underbrace{-1, \ldots, -1}_{\times q}), \tag{69}$$

2. $\mathrm{O}(p, q)$ *the* (pseudo-)orthogonal group *of signature* $(p, q)$:

$$\mathrm{O}(p, q) := \left\{ \Lambda \in \mathrm{GL}(\mathbb{R}^{p,q}) \,\middle|\, \Lambda^\top \Delta^{p,q} \Lambda = \Delta^{p,q} \right\}, \tag{70}$$

3. $\mathrm{SO}(p, q)$ *the* special (pseudo-)orthogonal group *of signature* $(p, q)$:

$$\mathrm{SO}(p, q) := \left\{ \Lambda \in \mathrm{O}(p, q) \,|\, \det \Lambda = 1 \right\}. \tag{71}$$

4. $\mathrm{O}^0(p, q) = \mathrm{SO}^0(p, q)$ *the identity component (=connected component of the identity) of* $\mathrm{O}(p, q)$. *Note that we have the inclusions of groups:*

$$\mathrm{O}^0(p, q) \subseteq \mathrm{SO}(p, q) \subseteq \mathrm{O}(p, q). \tag{72}$$

**Lemma D.1.5** (The connected components of $\mathrm{O}(p, q)$). *Let $p, q \geq 0$. Then we have the following:*

1. *If $p = q = 0$ then $\mathrm{O}(p, q) = \{0\}$ has only one connected component.*

2. *If $p + q \geq 1$ and $pq = 0$ then $\mathrm{O}(p, q)$ has 2 connected components:*

$$\mathrm{SO}(p, q) = \{\Lambda \in \mathrm{O}(p, q) \,|\, \det \Lambda = 1\} = \mathrm{O}^0(p, q), \quad \{\Lambda \in \mathrm{O}(p, q) \,|\, \det \Lambda = -1\}. \tag{73}$$

3. *If $p, q \geq 1$ then $\mathrm{O}(p, q)$ has 4 connected components:*

$$\mathrm{O}^{++}(p, q) := \left\{ \Lambda = \begin{bmatrix} A & B \\ C & D \end{bmatrix} \in \mathrm{O}(p, q) \,\middle|\, \det A \geq +1, \det D \geq +1 \right\} = \mathrm{O}^0(p, q), \tag{74}$$

$$\mathrm{O}^{+-}(p, q) := \left\{ \Lambda = \begin{bmatrix} A & B \\ C & D \end{bmatrix} \in \mathrm{O}(p, q) \,\middle|\, \det A \geq +1, \det D \leq -1 \right\}, \tag{75}$$

$$\mathrm{O}^{-+}(p, q) := \left\{ \Lambda = \begin{bmatrix} A & B \\ C & D \end{bmatrix} \in \mathrm{O}(p, q) \,\middle|\, \det A \leq -1, \det D \geq +1 \right\}, \tag{76}$$

$$\mathrm{O}^{--}(p, q) := \left\{ \Lambda = \begin{bmatrix} A & B \\ C & D \end{bmatrix} \in \mathrm{O}(p, q) \,\middle|\, \det A \leq -1, \det D \leq -1 \right\}. \tag{77}$$

*We have the following open-and-closed subgroups of $\mathrm{O}(p, q)$:*

$$\mathrm{O}^+(p, q) := \left\{ \Lambda = \begin{bmatrix} A & B \\ C & D \end{bmatrix} \in \mathrm{O}(p, q) \,\middle|\, \det A \geq +1 \right\} \tag{78}$$

$$= \mathrm{O}^{++}(p, q) \,\dot\cup\, \mathrm{O}^{+-}(p, q), \tag{79}$$

$$\mathrm{O}^-(p, q) := \left\{ \Lambda = \begin{bmatrix} A & B \\ C & D \end{bmatrix} \in \mathrm{O}(p, q) \,\middle|\, \det D \geq +1 \right\} \tag{80}$$

$$= \mathrm{O}^{++}(p, q) \,\dot\cup\, \mathrm{O}^{-+}(p, q), \tag{81}$$

$$\mathrm{SO}(p, q) = \{\Lambda \in \mathrm{O}(p, q) \,|\, \det \Lambda = 1\} \tag{82}$$

$$= \mathrm{O}^{++}(p, q) \,\dot\cup\, \mathrm{O}^{--}(p, q), \tag{83}$$

$$\mathrm{O}^0(p, q) = \mathrm{O}^+(p, q) \cap \mathrm{O}^-(p, q) \tag{84}$$

$$= \mathrm{O}^+(p, q) \cap \mathrm{SO}(p, q) \tag{85}$$

$$= \mathrm{O}^-(p, q) \cap \mathrm{SO}(p, q). \tag{86}$$

*In particular, if $p + q \geq 1$ and $p$ or $q$ is an odd number then $-I \notin \mathrm{O}^0(p, q)$.*

**Lemma D.1.6** (Polar representation). *Let $p, q \geq 1$. Then every $\Lambda \in \mathrm{O}(p, q)$ can uniquely be written as:*

$$\Lambda = \begin{bmatrix} U & 0 \\ 0 & V \end{bmatrix} \begin{bmatrix} \sqrt{I_p + BB^\top} & B \\ B^\top & \sqrt{I_q + B^\top B} \end{bmatrix}, \tag{87}$$

*with unique $U \in \mathrm{O}(p)$, $V \in \mathrm{O}(q)$, $B \in \mathbb{R}^{p \times q}$. Conversely, every such product of matrices lies in $\mathrm{O}(p, q)$.*

**Example D.1.7** (Affine conformal diffeomorphisms). *Let $b \in \mathbb{R}^{p,q}$, $c > 0$ and $\Lambda \in \mathrm{O}(p,q)$. Consider the affine map:*

$$f : \mathbb{R}^{p,q} \to \mathbb{R}^{p,q}, \qquad\qquad f(x) := c\Lambda x + b. \tag{88}$$

*Then $f$ is a conformal diffeomorphism of $\mathbb{R}^{p,q}$ with constant conformal factor $\omega(x) = c$.*

*Proof.* The differential at $x$ is given by the matrix:

$$df_x = c\Lambda. \tag{89}$$

So for $v_1, v_2 \in \mathrm{T}_x \mathbb{R}^{p,q} = \mathbb{R}^{p,q}$ we get:

$$\eta^{p,q}(df_x(v_1), df_x(v_2)) = (c\Lambda v_1)^\top \Delta^{p,q}(c\Lambda v_2) \tag{90}$$
$$= c^2 \cdot v_1^\top \Lambda^\top \Delta^{p,q} \Lambda v_2 \tag{91}$$
$$= c^2 \cdot v_1^\top \Delta^{p,q} v_2 \tag{92}$$
$$= c^2 \cdot \eta^{p,q}(v_1, v_2). \tag{93}$$

So, if we then define the conformal factor $\omega : \mathbb{R}^{p,q} \to \mathbb{R}_{>0}$ to be the constant map $\omega(x) := c$, we have shown that $f$ is a conformal smooth map. Its inverse $f^{-1}$ is given by:

$$f^{-1}(x) := c^{-1}\Lambda^{-1}x - c^{-1}\Lambda^{-1}b, \tag{94}$$

which is thus of the same form as $f$ and thus also a conformal smooth map. This shows the claim. $\square$

**Theorem D.1.8** (Affine conformal diffeomorphisms, see Amir-Moéz (1967)). *Consider the affine map on $\mathbb{R}^{p,q}$:*

$$f : \mathbb{R}^{p,q} \to \mathbb{R}^{p,q}, \qquad\qquad f(x) := Ax + b, \tag{95}$$

*with a square matrix $A$ and translation vector $b \in \mathbb{R}^{p,q}$. Then $f$ is a conformal map (w.r.t. $\eta^{p,q}$) iff there exists a $c \in \mathbb{R}_{>0}$ and a $\Lambda \in \mathrm{O}(p,q)$ such that $A = c\Lambda$. If this is the case, $f$ is a conformal diffeomorphism and both $c > 0$ and $\Lambda \in \mathrm{O}(p,q)$ are uniquely determined by $A$ as: $c = \sqrt[d]{|\det A|}$ with $d := p + q$, and, $\Lambda = \frac{1}{c}A$.*

*Proof.* One direction is proven in Example D.1.7. For the other direction assume that $f$ is a conformal map. Then we have $df_x = A$ and the conformal relation (in matrix form):

$$A^\top \Delta^{p,q} A = \omega(x)^2 \cdot \Delta^{p,q}. \tag{96}$$

Taking determinants on both sides gives:

$$(\det A)^2 = (\omega(x)^2)^d > 0, \tag{97}$$

showing that $A$ is invertible and that $\omega(x) = \sqrt[d]{|\det A|} =: c$ is not dependent on $x$ and thus equal to the constant $c > 0$. Dividing the first equation on both sides by $c^2$ and rearranging gives:

$$\left(\frac{1}{c}A\right)^\top \Delta^{p,q} \left(\frac{1}{c}A\right) = \Delta^{p,q}. \tag{98}$$

This shows that $\Lambda := \frac{1}{c}A \in \mathrm{O}(p,q)$ and thus the claim: $A = c\Lambda$. The rest follows from Example D.1.7. $\square$

**Definition D.1.9** (The linear and affine conformal group). *We define the* linear and affine conformal group, *resp., of signature $(p,q)$ as follows:*

$$\mathrm{CO}(p,q) := \mathbb{R}_{>0} \times \mathrm{O}(p,q), \qquad\qquad \mathrm{CE}(p,q) := \mathrm{CO}(p,q) \rtimes \mathbb{R}^{p,q}. \tag{99}$$

*Note that the entries correspond to scaling factor $c > 0$, reflection-rotation matrix $\Lambda \in \mathrm{O}(p,q)$ and translation vector $b \in \mathbb{R}^{p,q}$ in the conformal affine maps from Example D.1.7 and Theorem D.1.8.*

*We also define their identity components:*

$$\mathrm{CO}^0(p,q) := \mathbb{R}_{>0} \times \mathrm{O}^0(p,q), \qquad\qquad \mathrm{CE}^0(p,q) := \mathrm{CO}^0(p,q) \rtimes \mathbb{R}^{p,q}. \tag{100}$$

**Example D.1.10** (The inversion at the pseudo-sphere). *The following (partial) map:*

$$\varsigma = \varsigma^{p,q} : \mathbb{R}^{p,q} \to \mathbb{R}^{p,q}, \qquad\qquad \varsigma^{p,q}(x) := \frac{x}{\eta^{p,q}(x,x)}, \qquad (101)$$

*is called the* inversion (at the pseudo-sphere) *of signature* $(p,q)$. *Note that the inversion here is only defined for* $x \in U := \{\tilde{x} \in \mathbb{R}^{p,q} \mid \eta^{p,q}(\tilde{x}, \tilde{x}) \neq 0\}$. *It is an involution (self-invers) on* $U$ *as (with* $\eta := \eta^{p,q}$):

$$\eta(\varsigma(x), \varsigma(x)) = \frac{\eta(x,x)}{\eta(x,x)^2} = \frac{1}{\eta(x,x)} \neq 0, \qquad (102)$$

$$\varsigma(\varsigma(x)) = \frac{\varsigma(x)}{\eta(\varsigma(x), \varsigma(x))} = \frac{\frac{x}{\eta(x,x)}}{\frac{1}{\eta(x,x)}} = x. \qquad (103)$$

*We now claim that* $\varsigma : U \to U$ *is a conformal diffeomorphism. For this we compute its Jacobian matrix (differential), which, in matrix form, is given as:*

$$d\varsigma_x = \frac{1}{\eta(x,x)^2} \left( \eta(x,x)I - 2xx^{\top}\Delta^{p,q} \right). \qquad (104)$$

*To check the conformal relation we thus compute:*

$$(d\varsigma_x)^{\top}\Delta^{p,q}(d\varsigma_x) = \frac{1}{\eta(x,x)^4} \left( \left( \eta(x,x)I - 2xx^{\top}\Delta^{p,q} \right)^{\top} \Delta^{p,q} \left( \eta(x,x)I - 2xx^{\top}\Delta^{p,q} \right) \right) \quad (105)$$

$$= \frac{1}{\eta(x,x)^4} \left( \left( \eta(x,x)I - 2\Delta^{p,q}xx^{\top} \right) \Delta^{p,q} \left( \eta(x,x)I - 2xx^{\top}\Delta^{p,q} \right) \right) \qquad (106)$$

$$= \frac{1}{\eta(x,x)^4} \left( \eta(x,x)^2\Delta^{p,q} - 4\eta(x,x)\Delta^{p,q}xx^{\top}\Delta^{p,q} + 4\Delta^{p,q}xx^{\top}\Delta^{p,q}xx^{\top}\Delta^{p,q} \right)$$
$$(107)$$

$$= \frac{1}{\eta(x,x)^4} \left( \eta(x,x)^2\Delta^{p,q} - 4\eta(x,x)\Delta^{p,q}xx^{\top}\Delta^{p,q} + 4\eta(x,x)\Delta^{p,q}xx^{\top}\Delta^{p,q} \right)$$
$$(108)$$

$$= \frac{1}{\eta(x,x)^2}\Delta^{p,q}. \qquad (109)$$

*This shows that the inversion* $\varsigma : U \to U$ *is a conformal diffeomorphism with conformal factor* $\omega(x) = \frac{1}{|\eta(x,x)|}$. *We continue the analysis in Example D.4.5.*

**Example D.1.11** (Special conformal transformations). *For* $b \in \mathbb{R}^{p,q}$ *we define the (partial) map:*

$$\sigma_b : \mathbb{R}^{p,q} \to \mathbb{R}^{p,q}, \qquad \sigma_b(x) := \frac{x - \eta(x,x) \cdot b}{1 - 2 \cdot \eta(x,b) + \eta(b,b) \cdot \eta(x,x)} = \varsigma(\varsigma(x) - b), \qquad (110)$$

*where* $\varsigma = \varsigma^{p,q} : \mathbb{R}^{p,q} \to \mathbb{R}^{p,q}$ *denotes the inversion from Example D.1.10 of signature* $(p,q)$. *Maps of the form* $\sigma_b$ *are called* special conformal transformations *of* $\mathbb{R}^{p,q}$ *and are conformal maps on their domain of definition:*

$$U_b := \{x \in \mathbb{R}^{p,q} \mid \nu(x,b) \neq 0\}, \qquad (111)$$

*with conformal factor* $\omega_b(x) := \frac{1}{|\nu(x,b)|}$, *where we abbreviated the above denominator as:*

$$\nu(x,b) := 1 - 2 \cdot \eta(x,b) + \eta(b,b) \cdot \eta(x,x). \qquad (112)$$

*Furthermore,* $\sigma_{-b} : U_{-b} \to U_b$ *is the inverse of* $\sigma_b : U_b \to U_{-b}$.

*Proof.* We first check that $y := \sigma_b(x)$ lies in $U_{-b}$ for $x \in U_b$. For this we compute:

$$\eta(x - \eta(x,x)b, x - \eta(x,x)b) = \eta(x,x) - 2\eta(x,x)\eta(x,b) + \eta(x,x)^2\eta(b,b) \qquad (113)$$

$$= \eta(x,x)\nu(x,b), \qquad (114)$$

$$\eta(x - \eta(x,x)b, -b) = -\eta(x,b) + \eta(x,x)\eta(b,b) \qquad (115)$$

Dividing both sides by $\nu(x,b)^2$ and $\nu(x,b)$, resp., shows:

$$\eta(y,y) = \frac{\eta(x,x)}{\nu(x,b)}, \qquad \eta(y,-b) = \frac{1}{\nu(x,b)}\left(-\eta(x,b) + \eta(x,x)\eta(b,b)\right). \tag{116}$$

With this we get:

$$\nu(y,-b) = 1 - 2 \cdot \eta(y,-b) + \eta(b,b) \cdot \eta(y,y) \tag{117}$$

$$= \frac{1}{\nu(x,b)}\left(\nu(x,b) + 2 \cdot \eta(x,b) - 2 \cdot \eta(x,x) \cdot \eta(b,b) + \eta(b,b) \cdot \eta(x,x)\right) \tag{118}$$

$$= \frac{1}{\nu(x,b)}\left(\nu(x,b) + 2 \cdot \eta(x,b) - \eta(b,b) \cdot \eta(x,x)\right) \tag{119}$$

$$= \frac{1}{\nu(x,b)} \tag{120}$$

$$\neq 0. \tag{121}$$

This shows: $y \in U_{-b}$ for $x \in U_b$. Plugging the relation $\nu(y,-b) = \frac{1}{\nu(x,b)}$ into Equation (116) shows that:

$$\sigma_{-b}(y) = \frac{y + \eta(y,y) \cdot b}{\nu(y,-b)} \tag{122}$$

$$= \frac{1}{\nu(y,-b)} \cdot y + \frac{\eta(y,y)}{\nu(y,-b)} \cdot b \tag{123}$$

$$= \nu(x,b) \cdot \left(\frac{x - \eta(x,x) \cdot b}{\nu(x,b)}\right) + \eta(x,x) \cdot b \tag{124}$$

$$= x. \tag{125}$$

This shows that $\sigma_{-b} : U_{-b} \to U_b$ is the inverse of $\sigma_b : U_b \to U_{-b}$.

To show that $\sigma_b : U_b \to \mathbb{R}^{p,q}$ is a conformal map we need to compute its differential. Computing the differential directly gives:

$$(d\sigma_b)_x = \frac{\nu(x,b) \cdot \left(I - 2 \cdot b \cdot x^\top \Delta^{p,q}\right) - (x - \eta(x,x) \cdot b) \cdot \left(-2 \cdot b^\top \Delta^{p,q} + 2 \cdot \eta(b,b) \cdot x^\top \Delta^{p,q}\right)}{\nu(x,b)^2}, \tag{126}$$

which is difficult to work with. Instead, we use the fact that $\sigma_b$ is a smooth extension of the map $\sigma_b(x) = \varsigma\left(\varsigma(x) - b\right)$. With the chain rule we then get:

$$(d\sigma_b)_x = (d\varsigma)_{(\varsigma(x)-b)} \, d\varsigma_x \tag{127}$$

With this and Example D.1.10 we get:

$$(d\sigma_b)_x^\top \Delta^{p,q}(d\sigma_b)_x = (d\varsigma_x)^\top (d\varsigma)_{(\varsigma(x)-b)}^\top \Delta^{p,q}(d\varsigma)_{(\varsigma(x)-b)} \, d\varsigma_x \tag{128}$$

$$= \frac{1}{\eta\left(\varsigma(x)-b,\varsigma(x)-b\right)^2} \cdot (d\varsigma_x)^\top \Delta^{p,q} \, d\varsigma_x \tag{129}$$

$$= \frac{1}{\eta(x,x)^2} \cdot \frac{1}{\eta\left(\varsigma(x)-b,\varsigma(x)-b\right)^2} \Delta^{p,q} \tag{130}$$

$$= \frac{1}{\nu(x,b)^2} \Delta^{p,q}. \tag{131}$$

For the last step note that:

$$\eta\left(\varsigma(x)-b,\varsigma(x)-b\right) = \frac{\nu(x,b)}{\eta(x,x)}. \tag{132}$$

So the conformal factor of $\sigma_b$ at $x \in U_b$ is $\omega_b(x) := \frac{1}{|\nu(x,b)|}$. We continue the analysis in Example D.4.9. $\qquad\square$

**Remark D.1.12** (Extending conformal maps to a conformal compactification). *The examples D.1.7, D.1.8, D.1.10, D.1.11 have given us several conformal transformations of $\mathbb{R}^{p,q}$. However, for us to be able to define the inversion $\varsigma : U \to \mathbb{R}^{p,q}$ and the special conformal transformations $\sigma_b : U_b \to \mathbb{R}^{p,q}$ we had to restrict their domain of definition to an open subset of $\mathbb{R}^{p,q}$. The question is now if we can actually extend those maps like $\varsigma$ and $\sigma_b$ to a conformal map on the whole of $\mathbb{R}^{p,q}$ by possibly enlarging their codomain, e.g. by putting:*

$$\varsigma : \mathbb{R}^{p,q} \to \mathbb{R}^{p,q} \cup \{\infty\}, \qquad \varsigma(x) := \begin{cases} \frac{x}{\eta(x,x)}, & \text{if } \eta(x,x) \neq 0, \\ \infty, & \text{if } \eta(x,x) = 0 \, ? \end{cases} \tag{133}$$

*However, for this we would need to turn the extended codomain $\mathbb{R}^{p,q} \cup \{\infty\}$ into a proper pseudo-Riemannian manifold $\mathbb{M}^{p,q}$. Furthermore, we then would also like to properly define $\varsigma$ on the added new points $\mathbb{M}^{p,q} \setminus \mathbb{R}^{p,q}$ and turn it into a conformal diffeomorphism of $\mathbb{M}^{p,q}$. Note that in the non-Euclidean case, i.e. if $p, q > 0$, the space $\{x \in \mathbb{R}^{p,q} \mid \eta(x,x) = 0\}$ consists of more than one point and thus needs more consideration.*

*The above is the main question (of the existence) of a* conformal compactification *of $\mathbb{R}^{p,q}$. Below we follow an ad hoc approach to this question.*

**Remark D.1.13.** *Even after the question of conformal compactification, as described in Remark D.1.12, is solved, we are still faced with the ambiguity of how to define the* conformal group *of $\mathbb{R}^{p,q}$. There are several non-equivalent options:*

1. *as the group $\mathrm{ConfDiff}(\mathbb{R}^{p,q})$ of all conformal diffeomorphisms of $\mathbb{R}^{p,q}$, which would include all affine conformal transformations, but exclude the inversion and the special conformal transformations;*

2. *as the group $\mathrm{ConfDiff}(\mathbb{M}^{p,q})$ of all conformal diffeomorphisms of a conformal compactification $\mathbb{M}^{p,q}$ of $\mathbb{R}^{p,q}$, which would include all affine and special conformal transformations and the inversion. However, one can show that for $p + q = 2$ some of the properties of the conformal compactification will break down and that $\mathrm{ConfDiff}(\mathbb{M}^{p,q})$ can become pathologically big in the case $(p,q) = (1,1)$, see Schottenloher (2008);*

3. *as the connected component of the identity of one of the above groups, in order to stick to the main conformal diffeomorphisms that have descriptions with help of a Lie algebra and Lie exponential map, etc.;*

4. *as the subgroup of (partially defined) conformal diffeomorphisms of $\mathbb{R}^{p,q}$ that is generated by all affine and special conformal transformations (excluding the inversion) of $\mathbb{R}^{p,q}$;*

5. *as the subgroup of conformal diffeomorphisms of $\mathbb{M}^{p,q}$ that is generated by all affine and special conformal transformations (excluding the inversion) of $\mathbb{R}^{p,q}$;*

*For this draft we will settle with one of the last definitions, which are mostly equivalent, and call it the* (restricted) conformal group *of $\mathbb{R}^{p,q}$ and denote it by $\mathrm{Conf}(\mathbb{R}^{p,q})$. For more details see further below.*

To define the conformal compactification of $\mathbb{R}^{p,q}$ we first need to introduce the bigger pseudo-Euclidean space $\mathbb{R}^{p+1,q+1}$ and the projective space $\mathbb{P}^{p+q+1}$.

## D.2 The Projective Space, de Sitter Space and Anti-de Sitter Space

**Definition D.2.1** (The projective space). *Let $V$ be a (real) finite dimensional vector space. We then define the* projective space associated with $V$ *as the space of equivalence classes:*

$$\mathbb{P}(V) := (V \setminus \{0\}) / \sim, \tag{134}$$

*where we define two non-zero vectors $x_1, x_2 \in V$ to be equivalent if they lie on the same straight line through the origin:*

$$x_1 \sim x_2 : \iff \exists c \in \mathbb{R} \setminus \{0\}. \, x_1 = c \cdot x_2. \tag{135}$$

*We then define the* standard projective space *of dimensions $d \geq 0$ as:*

$$\mathbb{P}^d := \mathbb{P}(\mathbb{R}^{d+1}). \tag{136}$$

*Note that $d$ denotes the dimension of $\mathbb{P}^d$, which thus requires the Euclidean space $\mathbb{R}^{d+1}$ to be of one dimension higher.*

**Remark D.2.2.** *W.r.t. to our pseudo-Euclidean space $\mathbb{R}^{p,q}$ we will in the following mostly consider the projective space of dimension $d + 1$, where $d := p + q$:*

$$\mathbb{P}^{d+1} := \mathbb{P}(\mathbb{R}^{p+1,q+1}). \tag{137}$$

*Elements of $\mathbb{P}^{d+1}$ will be either denoted by $[z]$ with $z \in \mathbb{R}^{p+1,q+1}$ with coordinates $z = (z^0, z^1, \ldots, z^d, z^{d+1})$ or directly as $[z] = [z^0 : z^1 : \cdots : z^d : z^{d+1}]$. Note that we endow $\mathbb{R}^{p+1,q+1}$ with the standard metric $\eta^{p+1,q+1}$ of the pseudo-Euclidean space of dimension $d + 2$ and signature $(p + 1, q + 1)$ and that we consider the coordinates $z^0, \ldots, z^p$ to belong to the $(+1)$-signature and the coordinates $z^{p+1}, \ldots, z^{d+1}$ to belong to the $(-1)$-signature.*

**Definition D.2.3** (de Sitter and anti-de Sitter space). *Let $(V, \eta)$ be a non-degenerate finite dimensional real quadratic vector space. Then we define the* de Sitter space *and the* anti-de Sitter space *and the* (projective) zero quadric *associated to $(V, \eta)$ as:*

$$\mathrm{dS}(V, \eta) := \{[z] \in \mathbb{P}(V) \mid \eta(z, z) > 0\}, \tag{138}$$

$$\mathrm{AdS}(V, \eta) := \{[z] \in \mathbb{P}(V) \mid \eta(z, z) < 0\}. \tag{139}$$

$$\mathbb{M}(V, \eta) := \{[z] \in \mathbb{P}(V) \mid \eta(z, z) = 0\}. \tag{140}$$

*Note that these are all well-defined subsets of $\mathbb{P}(V)$. The pseudo-Riemannian metrics on the first two spaces are induced by the following subspaces of $V$:*

$$Y_+ := \{y \in V \mid \eta(y, y) = 1\}, \qquad Y_- := \{y \in V \mid \eta(y, y) = -1\}, \tag{141}$$

*and the $2 : 1$ locally diffeomorphic surjective maps:*

$$\pi_+ : Y_+ \to \mathrm{dS}(V, \eta) \subseteq \mathbb{P}(V), \qquad \pm y \mapsto [y], \tag{142}$$

$$\pi_- : Y_- \to \mathrm{AdS}(V, \eta) \subseteq \mathbb{P}(V), \qquad \pm y \mapsto [y], \tag{143}$$

*where both $Y_+$ and $Y_-$ are endowed with the pull-back metrics $\eta^+$ and $\eta^-$ from $\eta$ of $V$ to those subspaces. More explicitely, the tangent spaces and local metrics are given as follows:*

$$\mathrm{T}_{[z]}\,\mathrm{dS}(V, \eta) = \{v \in V \mid \eta(z, v) = 0\}, \qquad \eta^{\mathrm{dS}(V,\eta)}_{[z]}(v_1, v_2) = \eta(v_1, v_2), \tag{144}$$

$$\mathrm{T}_{[z]}\,\mathrm{AdS}(V, \eta) = \{v \in V \mid \eta(z, v) = 0\}, \qquad \eta^{\mathrm{AdS}(V,\eta)}_{[z]}(v_1, v_2) = \eta(v_1, v_2). \tag{145}$$

*We define the* (standard) de Sitter space *and the* (standard) anti-de Sitter space *and* (standard) zero quadric *of signatures $(p, q)$, resp., as follows:*

$$\mathrm{dS}^{p,q} := \mathrm{dS}(\mathbb{R}^{p+1,q}, \eta^{p+1,q}), \tag{146}$$

$$\mathrm{AdS}^{p,q} := \mathrm{AdS}(\mathbb{R}^{p,q+1}, \eta^{p,q+1}), \tag{147}$$

$$\mathbb{M}^{p,q} := \mathbb{M}(\mathbb{R}^{p+1,q+1}, \eta^{p+1,q+1}). \tag{148}$$

*The pseudo-Riemannian metric on $\mathbb{M}^{p,q}$ will be defined later in Definition D.3.5.*

**Definition D.2.4** (Projective orthogonal group). *Let $(V, \eta)$ be a non-degenerate finite dimensional real quadratic vector space. Then the* projective orthogonal group *of $(V, \eta)$ is defined to be:*

$$\mathrm{PO}(V, \eta) := \mathrm{O}(V, \eta)/\{\pm \mathrm{id}_V\}. \tag{149}$$

*The* (standard) projective orthogonal group *of signature $(p, q)$ is:*

$$\mathrm{PO}(p, q) := \mathrm{O}(p, q)/\{\pm I\}. \tag{150}$$

*We denote the identity component of those groups as:*

$$\mathrm{PO}^0(V, \eta), \qquad and \qquad \mathrm{PO}^0(p, q). \tag{151}$$

**Lemma D.2.5.** *Let $(V, \eta)$ be a non-degenerate finite dimensional real quadratic vector space. Then the projective orthogonal group $\mathrm{PO}(V, \eta)$ acts on the spaces $\mathbb{P}(V, \eta)$, $\mathrm{dS}(V, \eta)$, $\mathrm{AdS}(V, \eta)$ and on $\mathbb{M}(V, \eta)$ via matrix multiplication:*

$$[\Lambda][z] := [\Lambda z], \tag{152}$$

*in a well-defined way.*

**Theorem D.2.6** (See McKay (2023) Thm. 7.5). *Let $(V, \eta)$ be a non-degenerate finite dimensional real quadratic vector space. Then the standard action of $\mathrm{PO}(V, \eta)$ on $\mathrm{dS}(V, \eta)$ and $\mathrm{AdS}(V, \eta)$ acts isometrically and we get the following identification with their isometry groups:*

$$\mathrm{Isom}(\mathrm{dS}(V, \eta)) \cong \mathrm{PO}(V, \eta), \qquad\qquad \mathrm{Isom}(\mathrm{AdS}(V, \eta)) \cong \mathrm{PO}(V, \eta). \tag{153}$$

**Remark D.2.7.** *For our pseudo-Euclidean space $\mathbb{R}^{p,q}$ we will use the projective zero quadric $\mathbb{M}^{p,q}$ as its conformal compactification, see later in Definition D.3.5. For this we need to consider the bigger pseudo-Euclidean vector space $(\mathbb{R}^{p+1,q+1}, \eta^{p+1,q+1})$, and, the projective space $\mathbb{P}^{d+1} = \mathbb{P}(\mathbb{R}^{p+1,q+1})$. Here we get the disjoint decomposition of $\mathbb{P}^{d+1}$ into anti-de Sitter space, conformal compactification and de Sitter space:*

$$\mathbb{P}^{d+1} = \mathrm{AdS}^{p+1,q} \,\dot\cup\, \mathbb{M}^{p,q} \,\dot\cup\, \mathrm{dS}^{p,q+1} . \tag{154}$$

*Note the different signatures on each of the spaces. With Theorem D.2.6 we thus get the identifications:*

$$\mathrm{Isom}(\mathrm{dS}^{p,q+1}) = \mathrm{PO}(p+1, q+1) = \mathrm{Isom}(\mathrm{AdS}^{p+1,q}). \tag{155}$$

*In the following, we further want to show that the action of $\mathrm{PO}(p+1, q+1)$ on $\mathbb{M}^{p,q}$ induces conformal diffeomorphisms of $\mathbb{M}^{p,q}$. This identification via $\mathrm{PO}(p+1, q+1)$ will thus induce a* correspondence *between (certain)* conformal diffeomorphisms *of the conformal compactification $\mathbb{M}^{p,q}$ of $\mathbb{R}^{p,q}$ and the* isometries *of the corresponding* $\mathrm{AdS}^{p+1,q}$-space (or, also the $\mathrm{dS}^{p,q+1}$-space).

## D.3 The Conformal Compactification of the Pseudo-Euclidean Space

**Notation/Lemma D.3.1** (The isometric embedding). *The following map:*

$$\iota : \mathbb{R}^{p,q} \to \mathbb{R}^{p+1,q+1}, \qquad\qquad \iota(x) := \begin{bmatrix} \frac{1-\eta^{p,q}(x,x)}{2} \\ x \\ \frac{1+\eta^{p,q}(x,x)}{2} \end{bmatrix} . \tag{156}$$

*is an isometric embedding and satisfies for all $x \in \mathbb{R}^{p,q}$ :*

$$\iota(x) \neq 0, \qquad\qquad and \qquad\qquad \eta^{p+1,q+1}(\iota(x), \iota(x)) = 0. \tag{157}$$

*Proof.* For the latter consider the computation for $x \in \mathbb{R}^{p,q}$:

$$4\eta^{p+1,q+1}(\iota(x), \iota(x)) \tag{158}$$
$$= (1 - \eta^{p,q}(x,x))^2 + \eta^{p,q}(2x, 2x) - (1 + \eta^{p,q}(x,x))^2 \tag{159}$$
$$= 1 - 2\eta^{p,q}(x,x) + \eta^{p,q}(x,x)^2 + 4\eta^{p,q}(x,x) - 1 - 2\eta^{p,q}(x,x) - \eta^{p,q}(x,x)^2 \tag{160}$$
$$= 0. \tag{161}$$

This shows: $\eta^{p+1,q+1}(\iota(x), \iota(x)) = 0$. $\iota(x) \neq 0$ is clear. To see that $\iota$ is an isometric embedding compute the differential (Jacobian matrix):

$$d\iota_x = \begin{bmatrix} -x^\top \Delta^{p,q} \\ I \\ x^\top \Delta^{p,q} \end{bmatrix} . \tag{162}$$

With this we get:

$$(d\iota_x)^\top \Delta^{p+1,q+1} d\iota_x = \begin{bmatrix} -\Delta^{p,q}x & I & \Delta^{p,q}x \end{bmatrix} \Delta^{p+1,q+1} \begin{bmatrix} -x^\top \Delta^{p,q} \\ I \\ x^\top \Delta^{p,q} \end{bmatrix} \tag{163}$$

$$= \begin{bmatrix} -\Delta^{p,q}x & \Delta^{p,q} & -\Delta^{p,q}x \end{bmatrix} \begin{bmatrix} -x^\top \Delta^{p,q} \\ I \\ x^\top \Delta^{p,q} \end{bmatrix} \tag{164}$$

$$= \Delta^{p,q}xx^\top \Delta^{p,q} + \Delta^{p,q} - \Delta^{p,q}xx^\top \Delta^{p,q} \tag{165}$$
$$= \Delta^{p,q}. \tag{166}$$

This shows that $\iota$ is an isometric embedding. $\qquad\square$

**Definitions/Notations D.3.2.** *For $p, q \geq 0$, $d := p + q$, we introduce the following notations:*

1. *We introduce the* affine zero quadric *as the following subspace $Y_0$ of $\mathbb{R}^{p+1,q+1}$ via:*

$$Y_0 := \left\{ y \in \mathbb{R}^{p+1,q+1} \setminus \{0\} \,\middle|\, \eta^{p+1,q+1}(y, y) = 0 \right\}, \tag{167}$$

*and endow the tangent spaces for $y \in Y_0$:*

$$\mathrm{T}_y Y_0 = \left\{ v \in \mathbb{R}^{p+1,q+1} \,\middle|\, \eta^{p+1,q+1}(y, v) = 0 \right\} \subseteq \mathbb{R}^{p+1,q+1}, \tag{168}$$

*with the pull-back metric $\eta^{Y_0}$ of $\eta^{p+1,q+1}$:*

$$\eta_y^{Y_0}(v_1, v_2) = \eta^{p+1,q+1}(v_1, v_2) \qquad \text{for } v_1, v_2 \in \mathrm{T}_y Y_0. \tag{169}$$

2. *We also introduce the* double sphere*:*

$$\mathbb{S}^{p,q} := \mathbb{S}^p \times \mathbb{S}^q := \left\{ y \in \mathbb{R}^{p+1,q+1} \,\middle|\, \sum_{i=0}^{p} |y^i|^2 = \sum_{j=p+1}^{d+1} |y^j|^2 = 1 \right\} \subseteq Y_0, \tag{170}$$

*and endow the tangent spaces for $y \in \mathbb{S}^{p,q}$:*

$$\mathrm{T}_y \mathbb{S}^{p,q} = \left\{ v \in \mathbb{R}^{p+1,q+1} \,\middle|\, \sum_{i=0}^{p} y^i \cdot v^i = \sum_{j=p+1}^{d+1} y^j \cdot v^j = 0 \right\} \subseteq \mathbb{R}^{p+1,q+1}. \tag{171}$$

*with the pull-back metric $\eta^{\mathbb{S}^{p,q}}$ of $\eta^{p+1,q+1}$:*

$$\eta_y^{\mathbb{S}^{p,q}}(v_1, v_2) = \eta^{p+1,q+1}(v_1, v_2) \qquad \text{for } v_1, v_2 \in \mathrm{T}_y \mathbb{S}^{p,q}. \tag{172}$$

*Note that we have the inclusions:*

$$\iota(\mathbb{R}^{p,q}) \subseteq Y_0 \subseteq \mathbb{R}^{p+1,q+1}, \qquad\qquad \mathbb{S}^{p,q} \subseteq Y_0 \subseteq \mathbb{R}^{p+1,q+1}. \tag{173}$$

**Notation/Lemma D.3.3.** *Consider the maps:*

$$\rho : Y_0 \to \mathbb{R}, \qquad\qquad \rho(y) := \frac{1}{\sqrt{\sum_{i=0}^{p} |y^i|^2}}, \tag{174}$$

$$\psi : Y_0 \to \mathbb{S}^{p,q}, \qquad\qquad \psi(y) := \rho(y) \cdot y. \tag{175}$$

*Then $\psi$ is a well-defined map and conformal with conformal factor $\omega_\psi(y) = \rho(y)$.*

*Proof.* First note that for $y \in Y_0$ we have:

$$0 = \eta^{p+1,q+1}(y, y) = \sum_{i=0}^{p} |y^i|^2 - \sum_{j=p+1}^{d+1} |y^j|^2, \tag{176}$$

and thus:

$$\rho(y) = \frac{1}{\sqrt{\sum_{i=0}^{p} |y^i|^2}} \overset{!}{=} \frac{1}{\sqrt{\sum_{j=p+1}^{d+1} |y^j|^2}}. \tag{177}$$

This shows that for $y \in Y_0$ we have:

$$\psi(y) = \rho(y) \cdot y \in \mathbb{S}^p \times \mathbb{S}^q = \mathbb{S}^{p,q}. \tag{178}$$

The differential of $\psi$ at $y \in Y_0$:

$$d\psi_y : \mathrm{T}_y Y_0 \to \mathrm{T}_{\psi(y)} \mathbb{S}^{p,q} \subseteq \mathrm{T}_{\psi(y)} \mathbb{R}^{p+1,q+1}. \tag{179}$$

is given with help of the product rule as the matrix:

$$d\psi_y = y \cdot \rho'(y) + \rho(y) \cdot I. \tag{180}$$

With this we get for all $y \in Y_0$ and $v_1, v_2 \in \mathrm{T}_y Y_0$:

$$v_1^\top (d\psi_y)^\top \Delta^{p+1,q+1} (d\psi_y) v_2 \tag{181}$$

$$= v_1^\top (y \cdot \rho'(y) + \rho(y) \cdot I)^\top \Delta^{p+1,q+1} (y \cdot \rho'(y) + \rho(y) \cdot I) v_2 \tag{182}$$

$$= \rho(y)^2 \cdot v_1^\top \Delta^{p+1,q+1} v_2 + v_1^\top \rho'(y)^\top \underbrace{y^\top \Delta^{p+1,q+1} y}_{=0,\ \text{as}\ y \in Y_0} \rho'(y) v_2 \tag{183}$$

$$+ \rho(y) \cdot v_1^\top \rho'(y)^\top \underbrace{y^\top \Delta^{p+1,q+1} v_2}_{=0,\ \text{as}\ v_2 \in \mathrm{T}_y Y_0} + \underbrace{v_1^\top \Delta^{p+1,q+1} y}_{=0,\ \text{as}\ v_1 \in \mathrm{T}_y Y_0} \cdot \rho(y) \cdot \rho'(y) v_2 \tag{184}$$

$$= \rho(y)^2 \cdot v_1^\top \Delta^{p+1,q+1} v_2. \tag{185}$$

This shows the conformal factor of $\rho(y)$. $\qquad\square$

**Notation/Lemma D.3.4.** *Consider the following map:*

$$\tau : \mathbb{R}^{p,q} \overset{\iota}{\longrightarrow} Y_0 \overset{\psi}{\longrightarrow} \mathbb{S}^{p,q}, \tag{186}$$

$$\tau(x) := \psi(\iota(x)) = \frac{1}{\sqrt{\frac{1}{4}|1 - \eta^{p,q}(x,x)|^2 + \sum_{i=1}^p |x^i|^2}} \cdot \begin{bmatrix} \frac{1 - \eta^{p,q}(x,x)}{2} \\ x \\ \frac{1 + \eta^{p,q}(x,x)}{2} \end{bmatrix}. \tag{187}$$

*Then $\tau$ is a well-defined conformal map with conformal factor:*

$$\rho(\iota(x)) = \frac{1}{\sqrt{\frac{1}{4}|1 - \eta^{p,q}(x,x)|^2 + \sum_{i=1}^p |x^i|^2}}. \tag{188}$$

**Definition D.3.5** (The standard conformal compactification of the pseudo-Euclidean space). *For $p, q \geq 0$, $d := p + q$, the (standard) conformal compactification of $(\mathbb{R}^{p,q}, \eta^{p,q})$, is defined to be the following projective zero quadric subspace of the $(d+1)$-dimensional (real) projective space:*

$$\mathbb{M}^{p,q} := \left\{ [z] = [z^0 : z^1 : \cdots : z^d : z^{d+1}] \in \mathbb{P}^{d+1} \, \middle| \, \eta^{p+1,q+1}(z,z) = 0 \right\} \subseteq \mathbb{P}^{d+1}. \tag{189}$$

*Its metric $\eta^{\mathbb{M}^{p,q}}$ is induced with help of the* double sphere*:*

$$\mathbb{S}^{p,q} := \mathbb{S}^p \times \mathbb{S}^q := \left\{ y \in \mathbb{R}^{p+1,q+1} \, \middle| \, \sum_{i=0}^p |y^i|^2 = \sum_{j=p+1}^{d+1} |y^j|^2 = 1 \right\} \subseteq \mathbb{R}^{p+1,q+1}, \tag{190}$$

*and via the $2:1$ locally diffeomorphic surjective map:*

$$\pi : \mathbb{S}^{p,q} \to \mathbb{M}^{p,q}, \qquad\qquad \pm y \mapsto [y]. \tag{191}$$

*Note that if $[z] \in \mathbb{M}^{p,q}$ then we can put:*

$$y := \psi(z) = \rho(z) \cdot z \in \mathbb{R}^{p+1,q+1}, \tag{192}$$

*leading to the following properties[13]:*

$$y \in \mathbb{S}^{p,q}, \qquad\qquad [y] = [z] \in \mathbb{M}^{p,q}. \tag{193}$$

*More explicitly, the tangent space of $\mathbb{M}^{p,q}$ at $[z]$ is given by:*

$$\mathrm{T}_{[z]} \mathbb{M}^{p,q} = \left\{ v \in \mathbb{R}^{p+1,q+1} \, \middle| \, \sum_{i=0}^p z^i \cdot v^i = \sum_{j=p+1}^{d+1} z^j \cdot v^j = 0 \right\} \tag{194}$$

$$= \left\{ v \in \mathbb{R}^{p+1,q+1} \, \middle| \, \sum_{i=0}^p y^i \cdot v^i = \sum_{j=p+1}^{d+1} y^j \cdot v^j = 0 \right\} \tag{195}$$

$$= \mathrm{T}_y \mathbb{S}^{p,q} \subseteq \mathrm{T}_y \mathbb{R}^{p+1,q+1} = \mathbb{R}^{p+1,q+1}. \tag{196}$$

*and the metric of $\mathbb{M}^{p,q}$ at $[z]$ for $v_1, v_2 \in \mathrm{T}_{[z]} \mathbb{M}^{p,q}$ by:*

$$\eta^{\mathbb{M}^{p,q}}_{[z]}(v_1, v_2) := \eta^{\mathbb{S}^{p,q}}_y(v_1, v_2) = \eta^{p+1,q+1}(v_1, v_2) = v_1^\top \Delta^{p+1,q+1} v_2. \tag{197}$$

---

[13]By also multiplying $z$ with $\mathrm{sgn}(z^j)$ for one index $j \in \{0, 1, \ldots, d, d+1\}$, e.g. $j = d+1$, we can, in addition, arrange that $y^j \geq 0$ for this one fixed index $j$. For example, if $q = 0$ then $\mathbb{S}^{p,q} = \mathbb{S}^p \times \{\pm 1\}$, and we can always arrange the representative $y$ to have: $y^{d+1} = +1$. In this case: $\mathbb{M}^{p,0} \cong \mathbb{S}^p \times \{+1\}$.

**Definition D.3.6** (The conformal embedding of the pseudo-Euclidean space into its conformal compactification)**.** *The* conformal embedding *of* $\mathbb{R}^{p,q}$ *into* $\mathbb{M}^{p,q}$ *is defined to be:*

$$[\iota] : \mathbb{R}^{p,q} \to \mathbb{M}^{p,q}, \qquad [\iota(x)] := \left[ \frac{1 - \eta^{p,q}(x,x)}{2} : x : \frac{1 + \eta^{p,q}(x,x)}{2} \right] \qquad (198)$$

$$= [1 - \eta^{p,q}(x,x) : 2x : 1 + \eta^{p,q}(x,x)]. \qquad (199)$$

*Note that this is a well-defined map, as by Notation/Lemma D.3.1 we have:*

$$\iota(x) \neq 0, \qquad \eta^{p+1,q+1}(\iota(x), \iota(x)) = 0. \qquad (200)$$

*Note that we can also factorize the conformal embedding as follows:*

$$[\iota] : \mathbb{R}^{p,q} \xrightarrow{\tau} \mathbb{S}^{p,q} \xrightarrow{\pi} \mathbb{M}^{p,q}. \qquad (201)$$

*leading to the identity:*

$$[\tau(x)] = [\iota(x)], \qquad (202)$$

*and showing that the conformal embedding $[\iota]$ is a conformal map with conformal factor:*

$$\omega_{[\iota]}(x) = \omega_\tau(x) = \rho(\iota(x)) = \frac{1}{\sqrt{\frac{1}{4}|1 - \eta^{p,q}(x,x)|^2 + \sum_{i=1}^{p} |x^i|^2}}. \qquad (203)$$

## D.4 CONFORMAL TRANSFORMATIONS OF THE CONFORMAL COMPACTIFICATION OF THE PSEUDO-EUCLIDEAN SPACE

**Proposition D.4.1** (See Schottenloher (2008) Thm. 2.6)**.** *Let $p, q \geq 0$. Then every $[\Lambda] \in \mathrm{PO}(p + 1, q + 1)$ acts as a conformal diffeomorphism on the conformal compactification $\mathbb{M}^{p,q}$ of $\mathbb{R}^{p,q}$ via matrix multiplication and with the conformal factor:*

$$\omega_{[\Lambda]}([z]) := \sqrt{\frac{\sum_{i=0}^{p} |z^i|^2}{\sum_{i=0}^{p} |(\Lambda z)^i|^2}}. \qquad (204)$$

*The inverse of $[\Lambda]$ is given by $[\Lambda^{-1}]$. Furthermore, exactly the two matrices $\pm\Lambda \in \mathrm{O}(p + 1, q + 1)$ induce the same conformal diffeomorphism on $\mathbb{M}^{p,q}$, thus inducing an embedding/inclusion of groups:*

$$\mathrm{PO}(p + 1, q + 1) \subseteq \mathrm{ConfDiff}(\mathbb{M}^{p,q}). \qquad (205)$$

*Proof.* For every $\Lambda \in \mathrm{O}(p + 1, q + 1)$ it is clear that $\Lambda|_{Y_0}$ is an isometric map from $Y_0$ to $Y_0$. Now define the map:

$$\psi_\Lambda : \mathbb{S}^{p,q} \to \mathbb{S}^{p,q}, \qquad \psi_\Lambda(y) := \psi(\Lambda y). \qquad (206)$$

As the composition:

$$\psi_\Lambda : \mathbb{S}^{p,q} \subseteq Y_0 \xrightarrow{\Lambda} Y_0 \xrightarrow{\psi} \mathbb{S}^{p,q}, \qquad (207)$$

of the isometry $\Lambda$ (with conformal factor 1) and the conformal map $\psi$ (with conformal factor $\rho(y)$) also $\psi_\Lambda$ is a conformal map with the conformal factor $\rho(\Lambda y)$.

If now $[z] \in \mathbb{M}^{p,q}$ then $y := \rho(z) \cdot z \in \mathbb{S}^{p,q}$. This then shows that:

$$\psi_{[\Lambda]} : \mathbb{M}^{p,q} \to \mathbb{M}^{p,q}, \qquad \psi_{[\Lambda]}([z]) := [\Lambda z] = [\psi_\Lambda(y)], \qquad (208)$$

is a conformal map with conformal factor:

$$\omega_{[\Lambda]}([z]) = \rho(\Lambda y) = \rho(\Lambda(\rho(z)z)) = \frac{\rho(\Lambda z)}{\rho(z)} = \sqrt{\frac{\sum_{i=0}^{p} |z^i|^2}{\sum_{i=0}^{p} |(\Lambda z)^i|^2}}. \qquad (209)$$

Since that action is induced through matrix multiplication it is clear that $[\Lambda^{-1}]$ induces the inverse map to $[\Lambda]$.

Now assume that $\Lambda_1$ and $\Lambda_2$ induce the same map on $\mathbb{M}^{p,q}$. Then for every $y \in \mathbb{S}^{p,q}$ we have: $[\psi_{\Lambda_1}(y)] = [\psi_{\Lambda_2}(y)]$. So $\psi_{\Lambda_1}(y) = \pm\psi_{\Lambda_2}(y)$. Since this holds for all $y \in \mathbb{S}^{p,q}$, we get: $\Lambda_1 = \pm\Lambda_2$. This shows the claim. $\square$

**Theorem D.4.2** (See Schottenloher (2008) Thm. 2.6, Thm. 2.9, Thm. 2.11)**.** *If either $p + q \geq 3$ or $(p, q) = (2, 0)$ then the inclusion from Proposition D.4.1 is already an isomorphism:*

$$\mathrm{PO}(p + 1, q + 1) \cong \mathrm{ConfDiff}(\mathbb{M}^{p,q}). \tag{210}$$

*This means, in those $(p, q)$-cases, that every conformal diffeomorphism $\varphi : \mathbb{M}^{p,q} \to \mathbb{M}^{p,q}$ is given by the matrix multiplication with an (up to sign) unique matrix $\pm\Lambda \in \mathrm{O}(p+1, q+1)$, i.e.: $\varphi([z]) = [\Lambda z]$ for all $[z] \in \mathbb{M}^{p,q}$.*

**Theorem D.4.3** (See Schottenloher (2008) Thm. 2.6, Thm. 2.9, Thm. 2.11)**.** *In addition to the isomorphism in Equation (210) in Theorem D.4.2 we get the following stronger statements:*

1. *Let $p + q \geq 3$, then for every conformal map $\varphi : U \to \mathbb{R}^{p,q}$, defined on any connected open subset $U \subseteq \mathbb{R}^{p,q}$, there exists an (up to sign) unique matrix $\pm\Lambda \in \mathrm{O}(p + 1, q + 1)$ such that the following diagram commutes:*

$$
\begin{array}{ccc}
U & \xrightarrow{\varphi} & \mathbb{R}^{p,q} \\
{\scriptstyle[\iota]}\downarrow & & \downarrow{\scriptstyle[\iota]} \\
\mathbb{M}^{p,q} & \xrightarrow{[\Lambda]} & \mathbb{M}^{p,q}.
\end{array}
\tag{211}
$$

   *i.e. for all $x \in U$ we have:*

$$[\iota \circ \varphi(x)] = [\Lambda \iota(x)]. \tag{212}$$

   *In particular, $\varphi$ is injective.*

2. *Let $(p, q) = (2, 0)$, then for every* injective *conformal map $\varphi : U \to \mathbb{R}^{2,0}$, defined either on $U = \mathbb{R}^{2,0}$ or on any punctured plane $U = \mathbb{R}^{2,0} \setminus \{\tilde{x}\}$, there exists an (up to sign) unique matrix $\pm\Lambda \in \mathrm{O}(3, 1)$ such that the corresponding diagram from 211 commutes.*[14]

**Remark D.4.4.** *For $(p, q) = (1, 1)$ the group $\mathrm{ConfDiff}(\mathbb{M}^{p,q})$ is much bigger than $\mathrm{PO}(p + 1, q + 1)$. For details see Schottenloher (2008) section 2.5.*

**Example D.4.5** (The inversion at the pseudo-sphere)**.** *We continue the discussion about the* inversion (at the pseudo-sphere) $\varsigma$ on $\mathbb{R}^{p,q}$ *from Example D.1.10:*

$$\varsigma = \varsigma^{p,q} : U \to \mathbb{R}^{p,q}, \qquad\qquad \varsigma^{p,q}(x) := \frac{x}{\eta^{p,q}(x, x)}. \tag{213}$$

*defined on $U := \{x \in \mathbb{R}^{p,q} \mid \eta^{p,q}(x, x) \neq 0\}$. Consider the $(d + 2) \times (d + 2)$-matrix:*

$$\Lambda_\varsigma := \begin{bmatrix} -1 & 0 & 0 \\ 0 & I & 0 \\ 0 & 0 & 1 \end{bmatrix}. \tag{214}$$

*We now claim that the map:*

$$\bar{\varsigma} := [\Lambda_\varsigma] : \mathbb{M}^{p,q} \to \mathbb{M}^{p,q}, \qquad\qquad [\Lambda_\varsigma][z] := [\Lambda_\varsigma z], \tag{215}$$

---

[14]Note that, for the case $(p, q) = (2, 0)$, the injectivity of $\varphi$ needs to be assumed and we also can only allow for open subsets $U$ of $\mathbb{R}^{2,0}$ where at most one point is removed from $\mathbb{R}^{2,0}$.

*is the conformal extension of $\varsigma$ from $U$ to whole $\mathbb{M}^{p,q}$. Indeed, first note that $\Lambda_\varsigma \in \mathrm{O}(p+1, q+1)$ and $\det \Lambda_\varsigma = -1$ and $\det(-\Lambda_\varsigma) = (-1)^{d+1}$. Then compute:*

$$\iota(\varsigma(x)) = \frac{1}{2} \begin{bmatrix} 1 - \eta(\varsigma(x), \varsigma(x)) \\ 2\varsigma(x) \\ 1 + \eta(\varsigma(x), \varsigma(x)) \end{bmatrix} \tag{216}$$

$$= \frac{1}{2} \begin{bmatrix} 1 - \frac{1}{\eta(x,x)} \\ \frac{2x}{\eta(x,x)} \\ 1 + \frac{1}{\eta(x,x)} \end{bmatrix} \tag{217}$$

$$= \frac{1}{2} \frac{1}{\eta(x,x)} \begin{bmatrix} \eta(x,x) - 1 \\ 2x \\ \eta(x,x) + 1 \end{bmatrix} \tag{218}$$

$$= \frac{1}{2} \frac{1}{\eta(x,x)} \begin{bmatrix} -1 & 0 & 0 \\ 0 & I & 0 \\ 0 & 0 & 1 \end{bmatrix} \begin{bmatrix} 1 - \eta(x,x) \\ 2x \\ 1 + \eta(x,x) \end{bmatrix} \tag{219}$$

$$= \frac{1}{\eta(x,x)} \cdot \Lambda_\varsigma \iota(x), \tag{220}$$

*which shows the claim:*

$$[\Lambda_\varsigma \iota(x)] = [\iota(\varsigma(x))] \in \mathbb{M}^{p,q}. \tag{221}$$

*Note that for $x \in \mathbb{R}^{p,q}$ with $\eta(x,x) = 0$ we get:*

$$\iota(x) = [1 : 2x : 1], \qquad\qquad \bar\varsigma(\iota(x)) = [-1 : 2x : 1], \tag{222}$$

$$\iota(0) = [1 : 0 : 1], \qquad\qquad \bar\varsigma(\iota(0)) = [-1 : 0 : 1]. \tag{223}$$

**Example D.4.6** (Linear conformal transformations, see Schottenloher (2008) Thm. 2.9). *We continue from Example D.1.7 and Theorem D.1.8 for the linear conformal map: $x \mapsto Ax$ with the matrix $A = c\Lambda$, where $A = c\Lambda \in \mathrm{CO}(p,q)$ with $c > 0$ and $\Lambda \in \mathrm{O}(p,q)$. Then we can define the $(d+2) \times (d+2)$-matrix:*

$$\Gamma_{c\Lambda} := \begin{bmatrix} \frac{1+c^2}{2c} & 0 & \frac{1-c^2}{2c} \\ 0 & \Lambda & 0 \\ \frac{1-c^2}{2c} & 0 & \frac{1+c^2}{2c} \end{bmatrix}. \tag{224}$$

*Then $\Gamma_{c\Lambda} \in \mathrm{O}(p+1, q+1)$ as:*

$$\Gamma_{c\Lambda}^\top \Delta^{p+1,q+1} \Gamma_{c\Lambda} = \begin{bmatrix} \frac{1+c^2}{2c} & 0 & \frac{1-c^2}{2c} \\ 0 & \Lambda^\top & 0 \\ \frac{1-c^2}{2c} & 0 & \frac{1+c^2}{2c} \end{bmatrix} \begin{bmatrix} 1 & 0 & 0 \\ 0 & \Delta^{p,q} & 0 \\ 0 & 0 & -1 \end{bmatrix} \begin{bmatrix} \frac{1+c^2}{2c} & 0 & \frac{1-c^2}{2c} \\ 0 & \Lambda & 0 \\ \frac{1-c^2}{2c} & 0 & \frac{1+c^2}{2c} \end{bmatrix} \tag{225}$$

$$= \begin{bmatrix} \frac{1+c^2}{2c} & 0 & \frac{1-c^2}{2c} \\ 0 & \Lambda^\top & 0 \\ \frac{1-c^2}{2c} & 0 & \frac{1+c^2}{2c} \end{bmatrix} \begin{bmatrix} \frac{1+c^2}{2c} & 0 & \frac{1-c^2}{2c} \\ 0 & \Delta^{p,q}\Lambda & 0 \\ -\frac{1-c^2}{2c} & 0 & -\frac{1+c^2}{2c} \end{bmatrix} \tag{226}$$

$$= \begin{bmatrix} \left(\frac{1+c^2}{2c}\right)^2 - \left(\frac{1-c^2}{2c}\right)^2 & 0 & 0 \\ 0 & \Lambda^\top \Delta^{p,q} \Lambda & 0 \\ 0 & 0 & \left(\frac{1-c^2}{2c}\right)^2 - \left(\frac{1+c^2}{2c}\right)^2 \end{bmatrix} \tag{227}$$

$$= \begin{bmatrix} 1 & 0 & 0 \\ 0 & \Delta^{p,q} & 0 \\ 0 & 0 & -1 \end{bmatrix} \tag{228}$$

$$= \Delta^{p+1,q+1}. \tag{229}$$

*Also note that if $\Lambda \in \mathrm{O}^0(p,q)$ then $\Gamma_{c\Lambda} \in \mathrm{O}^0(p+1, q+1)$ as $c \to 1$ provides a path to the identity component.*[15] *We now claim that:*

$$[\Gamma_{c\Lambda}] : \mathbb{M}^{p,q} \to \mathbb{M}^{p,q}, \qquad\qquad [\Gamma_{c\Lambda}][z] := [\Gamma_{c\Lambda} z], \tag{230}$$

---

[15]The reverse is also true: $\Gamma_{c\Lambda} \in \mathrm{O}^0(p+1, q+1) \implies \Lambda \in \mathrm{O}^0(p,q)$.

*is the conformal extension of $c\Lambda$ from $\mathbb{R}^{p,q}$ to $\mathbb{M}^{p,q}$.*

$$\Gamma_{c\Lambda}\iota(x) = \begin{bmatrix} \frac{1+c^2}{2c} & 0 & \frac{1-c^2}{2c} \\ 0 & \Lambda & 0 \\ \frac{1-c^2}{2c} & 0 & \frac{1+c^2}{2c} \end{bmatrix} \begin{bmatrix} \frac{1-\eta^{p,q}(x,x)}{2} \\ x \\ \frac{1+\eta^{p,q}(x,x)}{2} \end{bmatrix} \tag{231}$$

$$= \begin{bmatrix} \left(\frac{1+c^2}{2c}\right)\left(\frac{1-\eta^{p,q}(x,x)}{2}\right) + \left(\frac{1-c^2}{2c}\right)\left(\frac{1+\eta^{p,q}(x,x)}{2}\right) \\ \Lambda x \\ \left(\frac{1-c^2}{2c}\right)\left(\frac{1-\eta^{p,q}(x,x)}{2}\right) + \left(\frac{1+c^2}{2c}\right)\left(\frac{1+\eta^{p,q}(x,x)}{2}\right) \end{bmatrix} \tag{232}$$

$$= \frac{1}{4c} \begin{bmatrix} 1 + c^2 - \eta^{p,q}(x,x) - c^2 \cdot \eta^{p,q}(x,x) + 1 - c^2 + \eta^{p,q}(x,x) - c^2 \cdot \eta^{p,q}(x,x) \\ 4c\Lambda x \\ 1 - c^2 - \eta^{p,q}(x,x) + c^2 \cdot \eta^{p,q}(x,x) + 1 + c^2 + \eta^{p,q}(x,x) + c^2 \cdot \eta^{p,q}(x,x) \end{bmatrix} \tag{233}$$

$$= \frac{1}{4c} \begin{bmatrix} 2 - 2 \cdot c^2 \cdot \eta^{p,q}(x,x) \\ 4c\Lambda x \\ 2 + 2 \cdot c^2 \cdot \eta^{p,q}(x,x) \end{bmatrix} \tag{234}$$

$$= \frac{1}{c} \begin{bmatrix} \frac{1-\eta^{p,q}(c\Lambda x, c\Lambda x)}{2} \\ c\Lambda x \\ \frac{1+\eta^{p,q}(c\Lambda x, c\Lambda x)}{2} \end{bmatrix} \tag{235}$$

$$= \frac{1}{c} \cdot \iota\left(c\Lambda x\right). \tag{236}$$

*This shows:*

$$[\Gamma_{c\Lambda}\iota(x)] = [\iota(c\Lambda x)] \in \mathbb{M}^{p,q}, \tag{237}$$

*and thus the claim.*

**Example D.4.7** (Translations, see Schottenloher (2008) Thm. 2.9)**.** *We now continue from Example D.1.7 and Theorem D.1.8 for case of translation: $x \mapsto x + b$ for $b \in \mathbb{R}^{p,q}$. Then we can define the $(d+2) \times (d+2)$-matrix:*

$$\Gamma_{I,b} := I + \Omega_b \tag{238}$$

$$:= I + \begin{bmatrix} -\frac{1}{2}\eta^{p,q}(b,b) & -b^\top \Delta^{p,q} & -\frac{1}{2}\eta^{p,q}(b,b) \\ b & 0 & b \\ \frac{1}{2}\eta^{p,q}(b,b) & b^\top \Delta^{p,q} & \frac{1}{2}\eta^{p,q}(b,b) \end{bmatrix} \tag{239}$$

$$= \begin{bmatrix} 1 - \frac{1}{2}\eta^{p,q}(b,b) & -b^\top \Delta^{p,q} & -\frac{1}{2}\eta^{p,q}(b,b) \\ b & I & b \\ \frac{1}{2}\eta^{p,q}(b,b) & b^\top \Delta^{p,q} & 1 + \frac{1}{2}\eta^{p,q}(b,b) \end{bmatrix}. \tag{240}$$

*We first show that $\Gamma_{I,b} \in \mathrm{O}(p+1, q+1)$. For this consider:*

$$\Gamma_{I,b}^\top \Delta^{p+1,q+1}\Gamma_{I,b} = \Delta^{p+1,q+1} + \Omega_b^\top \Delta^{p+1,q+1} + \Delta^{p+1,q+1}\Omega_b + \Omega_b^\top \Delta^{p+1,q+1}\Omega_b. \tag{241}$$

*We compute terms separately:*

$$\Delta^{p+1,q+1}\Omega_b = \begin{bmatrix} 1 & 0 & 0 \\ 0 & \Delta^{p,q} & 0 \\ 0 & 0 & -1 \end{bmatrix} \begin{bmatrix} -\frac{1}{2}\eta^{p,q}(b,b) & -b^\top \Delta^{p,q} & -\frac{1}{2}\eta^{p,q}(b,b) \\ b & 0 & b \\ \frac{1}{2}\eta^{p,q}(b,b) & b^\top \Delta^{p,q} & \frac{1}{2}\eta^{p,q}(b,b) \end{bmatrix} \tag{242}$$

$$= \begin{bmatrix} -\frac{1}{2}\eta^{p,q}(b,b) & -b^\top \Delta^{p,q} & -\frac{1}{2}\eta^{p,q}(b,b) \\ \Delta^{p,q}b & 0 & \Delta^{p,q}b \\ -\frac{1}{2}\eta^{p,q}(b,b) & -b^\top \Delta^{p,q} & -\frac{1}{2}\eta^{p,q}(b,b) \end{bmatrix}, \tag{243}$$

$$\Omega_b^\top \Delta^{p+1,q+1} = \left(\Delta^{p+1,q+1}\Omega_b\right)^\top \tag{244}$$

$$= \begin{bmatrix} -\frac{1}{2}\eta^{p,q}(b,b) & b^\top \Delta^{p,q} & -\frac{1}{2}\eta^{p,q}(b,b) \\ -\Delta^{p,q}b & 0 & -\Delta^{p,q}b \\ -\frac{1}{2}\eta^{p,q}(b,b) & b^\top \Delta^{p,q} & -\frac{1}{2}\eta^{p,q}(b,b) \end{bmatrix}. \tag{245}$$

*With this we compute:*

$$\Omega_b^\top \Delta^{p+1,q+1} + \Delta^{p+1,q+1}\Omega_b = -\eta^{p,q}(b,b) \cdot \begin{bmatrix} 1 & 0 & 1 \\ 0 & 0 & 0 \\ 1 & 0 & 1 \end{bmatrix}, \tag{246}$$

$$\tag{247}$$

*We also get:*

$$\Omega_b^\top \Delta^{p+1,q+1}\Omega_b = \begin{bmatrix} -\frac{1}{2}\eta^{p,q}(b,b) & b^\top \Delta^{p,q} & -\frac{1}{2}\eta^{p,q}(b,b) \\ -\Delta^{p,q}b & 0 & -\Delta^{p,q}b \\ -\frac{1}{2}\eta^{p,q}(b,b) & b^\top \Delta^{p,q} & -\frac{1}{2}\eta^{p,q}(b,b) \end{bmatrix} \begin{bmatrix} -\frac{1}{2}\eta^{p,q}(b,b) & -b^\top \Delta^{p,q} & -\frac{1}{2}\eta^{p,q}(b,b) \\ b & 0 & b \\ \frac{1}{2}\eta^{p,q}(b,b) & b^\top \Delta^{p,q} & \frac{1}{2}\eta^{p,q}(b,b) \end{bmatrix} \tag{248}$$

$$= \begin{bmatrix} b^\top \Delta^{p,q}b & 0 & b^\top \Delta^{p,q}b \\ 0 & 0 & 0 \\ b^\top \Delta^{p,q}b & 0 & b^\top \Delta^{p,q}b \end{bmatrix} \tag{249}$$

$$= \eta^{p,q}(b,b) \cdot \begin{bmatrix} 1 & 0 & 1 \\ 0 & 0 & 0 \\ 1 & 0 & 1 \end{bmatrix}. \tag{250}$$

*Together this shows:*

$$\Gamma_{I,b}^\top \Delta^{p+1,q+1}\Gamma_{I,b} = \Delta^{p+1,q+1} + \Omega_b^\top \Delta^{p+1,q+1} + \Delta^{p+1,q+1}\Omega_b + \Omega_b^\top \Delta^{p+1,q+1}\Omega_b \tag{251}$$

$$= \Delta^{p+1,q+1} - \eta^{p,q}(b,b) \cdot \begin{bmatrix} 1 & 0 & 1 \\ 0 & 0 & 0 \\ 1 & 0 & 1 \end{bmatrix} + \eta^{p,q}(b,b) \cdot \begin{bmatrix} 1 & 0 & 1 \\ 0 & 0 & 0 \\ 1 & 0 & 1 \end{bmatrix} \tag{252}$$

$$= \Delta^{p+1,q+1}. \tag{253}$$

*This thus shows the claim: $\Gamma_{I,b} \in \mathrm{O}(p+1,q+1)$. Furthermore, for $b \to 0$ we see that $\Gamma_{I,b} \to I$. This thus even shows that: $\Gamma_{I,b} \in \mathrm{O}^0(p+1,q+1)$.*

*We now claim that $[\Gamma_{I,b}]$ is the conformal extension of the translation map $x \mapsto x + b$ from $\mathbb{R}^{p,q}$ to $\mathbb{M}^{p,q}$. For this compute:*

$$\Gamma_{I,b}\iota(x) = (I + \Omega_b)\iota(x) \tag{254}$$
$$= \iota(x) + \Omega_b\iota(x), \tag{255}$$

*with:*

$$\Omega_b\iota(x) = \frac{1}{2} \begin{bmatrix} -\frac{1}{2}\eta^{p,q}(b,b) & -b^\top \Delta^{p,q} & -\frac{1}{2}\eta^{p,q}(b,b) \\ b & 0 & b \\ \frac{1}{2}\eta^{p,q}(b,b) & b^\top \Delta^{p,q} & \frac{1}{2}\eta^{p,q}(b,b) \end{bmatrix} \begin{bmatrix} 1 - \eta^{p,q}(x,x) \\ 2x \\ 1 + \eta^{p,q}(x,x) \end{bmatrix} \tag{256}$$

$$= \frac{1}{2} \begin{bmatrix} -\eta^{p,q}(b,b) - 2b^\top \Delta^{p,q}x \\ 2b \\ \eta^{p,q}(b,b) + 2b^\top \Delta^{p,q}x \end{bmatrix} \tag{257}$$

$$= \frac{1}{2} \begin{bmatrix} -\eta^{p,q}(b,b) - 2\eta^{p,q}(b,x) \\ 2b \\ \eta^{p,q}(b,b) + 2\eta^{p,q}(b,x) \end{bmatrix}. \tag{258}$$

*With this we get:*

$$\iota(x+b) = \frac{1}{2} \begin{bmatrix} 1 - \eta^{p,q}(x+b, x+b) \\ 2(x+b) \\ 1 + \eta^{p,q}(x+b, x+b) \end{bmatrix} \tag{259}$$

$$= \frac{1}{2} \begin{bmatrix} 1 - \eta^{p,q}(x,x) - \eta^{p,q}(b,b) - 2\eta^{p,q}(b,x) \\ 2x + 2b \\ 1 + \eta^{p,q}(x,x) + \eta^{p,q}(b,b) + 2\eta^{p,q}(b,x) \end{bmatrix} \tag{260}$$

$$= \frac{1}{2} \begin{bmatrix} 1 - \eta^{p,q}(x,x) \\ 2x \\ 1 + \eta^{p,q}(x,x) \end{bmatrix} + \frac{1}{2} \begin{bmatrix} -\eta^{p,q}(b,b) - 2\eta^{p,q}(b,x) \\ 2b \\ \eta^{p,q}(b,b) + 2\eta^{p,q}(b,x) \end{bmatrix} \tag{261}$$

$$= \iota(x) + \Omega_b\iota(x) \tag{262}$$
$$= \Gamma_{I,b}\iota(x). \tag{263}$$

*This then implies:*

$$[\Gamma_{I,b}\iota(x)] = [\iota(x+b)] \in \mathbb{M}^{p,q}. \tag{264}$$

*This shows that $[\Gamma_{I,b}]$ conformally extends the translation map $x \mapsto x+b$ from $\mathbb{R}^{p,q}$ to $\mathbb{M}^{p,q}$.*

**Example D.4.8** (Affine conformal transformations, see Schottenloher (2008) Thm. 2.9). *We continue from Example D.1.7 and Theorem D.1.8 for the affine conformal map: $x \mapsto Ax + b$ with the matrix $A = c\Lambda$ and $b \in \mathbb{R}^{p,q}$, where $A = c\Lambda \in \mathrm{CO}(p,q)$ with $c > 0$ and $\Lambda \in \mathrm{O}(p,q)$. Then we can define the $(d+2) \times (d+2)$-matrix:*

$$\Gamma_{c\Lambda,b} := \Gamma_{I,b}\Gamma_{c\Lambda} \tag{265}$$

$$= \begin{bmatrix} 1 - \frac{1}{2}\eta^{p,q}(b,b) & -b^\top\Delta^{p,q} & -\frac{1}{2}\eta^{p,q}(b,b) \\ b & I & b \\ \frac{1}{2}\eta^{p,q}(b,b) & b^\top\Delta^{p,q} & 1 + \frac{1}{2}\eta^{p,q}(b,b) \end{bmatrix} \begin{bmatrix} \frac{1+c^2}{2c} & 0 & \frac{1-c^2}{2c} \\ 0 & \Lambda & 0 \\ \frac{1-c^2}{2c} & 0 & \frac{1+c^2}{2c} \end{bmatrix}. \tag{266}$$

*Then we have: $\Gamma_{c\Lambda} \in \mathrm{O}(p+1,q+1)$. This is clear, as both matrices $\Gamma_{I,b}, \Gamma_{c\Lambda} \in \mathrm{O}(p+1,q+1)$ by Example D.4.6 and Example D.4.7. It thus also clear that the matrix $[\Gamma_{c\Lambda,b}]$ extends the affine map $x \mapsto c\Lambda x + b$ from $\mathbb{R}^{p,q}$ to $\mathbb{M}^{p,q}$.*

**Example D.4.9** (Special conformal transformations, see Schottenloher (2008) Thm. 2.9). *We continue from Example D.1.11 about the* special conformal transformations. *Recall that we defined the special conformal transformations for $b \in \mathbb{R}^{p,q}$ on an open subset $U_b \subseteq \mathbb{R}^{p,q}$ as:*

$$\sigma_b : U_b \to \mathbb{R}^{p,q}, \qquad \sigma_b(x) := \frac{x - \eta(x,x) \cdot b}{1 - 2 \cdot \eta(x,b) + \eta(b,b) \cdot \eta(x,x)} = \varsigma\left(\varsigma(x) - b\right), \tag{267}$$

*where $\varsigma$ is the inversion at the pseudo-sphere from Example D.1.10 and Example D.4.5. With the latter representation and Example D.4.7 we immediately get that the $(d+2) \times (d+2)$-matrix:*

$$\Sigma_b := \Lambda_\varsigma \Gamma_{I,-b} \Lambda_\varsigma \tag{268}$$

$$= \begin{bmatrix} -1 & 0 & 0 \\ 0 & I & 0 \\ 0 & 0 & 1 \end{bmatrix} \begin{bmatrix} 1 - \frac{1}{2}\eta^{p,q}(b,b) & b^\top\Delta^{p,q} & -\frac{1}{2}\eta^{p,q}(b,b) \\ -b & I & -b \\ \frac{1}{2}\eta^{p,q}(b,b) & -b^\top\Delta^{p,q} & 1 + \frac{1}{2}\eta^{p,q}(b,b) \end{bmatrix} \begin{bmatrix} -1 & 0 & 0 \\ 0 & I & 0 \\ 0 & 0 & 1 \end{bmatrix} \tag{269}$$

$$= \begin{bmatrix} 1 - \frac{1}{2}\eta^{p,q}(b,b) & -b^\top\Delta^{p,q} & \frac{1}{2}\eta^{p,q}(b,b) \\ b & I & -b \\ -\frac{1}{2}\eta^{p,q}(b,b) & -b^\top\Delta^{p,q} & 1 + \frac{1}{2}\eta^{p,q}(b,b) \end{bmatrix} \tag{270}$$

*lies in $\mathrm{O}(p+1,q+1)$ and that $[\Sigma_b]$ extends the special conformal transformation $\sigma_b$ from $\mathbb{R}^{p,q}$ to $\mathbb{M}^{p,q}$. Furthermore, for $b \to 0$ we see that $\Sigma_b \to I$. This then even shows that: $\Sigma_b \in \mathrm{O}^0(p+1,q+1)$.*

## D.5  THE DEFINITION OF THE CONFORMAL GROUP OF THE PSEUDO-EUCLIDEAN SPACE

**Definition D.5.1** (The global and restricted conformal group of the pseudo-Euclidean space). *Based on Theorem D.4.2 we define for all $p, q \geq 0$ the* global conformal group *of $\mathbb{R}^{p,q}$ to be:*

$$\mathrm{Conf}_g(\mathbb{R}^{p,q}) := \mathrm{PO}(p+1,q+1), \tag{271}$$

*and the* (restricted) conformal group $\mathrm{Conf}(\mathbb{R}^{p,q})$ *of $\mathbb{R}^{p,q}$ to be the connected component of the identity of $\mathrm{PO}(p+1,q+1)$:*

$$\mathrm{Conf}(\mathbb{R}^{p,q}) := \mathrm{Conf}_r(\mathbb{R}^{p,q}) := \mathrm{Conf}_g^0(\mathbb{R}^{p,q}) = \mathrm{PO}^0(p+1,q+1). \tag{272}$$

*Note that by Proposition D.4.1 we always have the inclusions:*

$$\mathrm{Conf}_r(\mathbb{R}^{p,q}) \subseteq \mathrm{Conf}_g(\mathbb{R}^{p,q}) \subseteq \mathrm{ConfDiff}(\mathbb{M}^{p,q}). \tag{273}$$

**Theorem D.5.2.** $\mathrm{Conf}(\mathbb{R}^{p,q})$ *contains all affine conformal transformations $x \mapsto Ax + b$ of $\mathbb{R}^{p,q}$ where $A = c\Lambda \in \mathrm{CO}^0(p,q)$ with $c > 0$ and $\Lambda \in \mathrm{O}^0(p,q)$ and $b \in \mathbb{R}^{p,q}$ in form of the map $[\Gamma_{c\Lambda,b}] : \mathbb{M}^{p,q} \to \mathbb{M}^{p,q}$ from Example D.4.8, and, also all the special conformal transformations $\sigma_b$ in form of the map $[\Sigma_b] : \mathbb{M}^{p,q} \to \mathbb{M}^{p,q}$ from Example D.4.9.*

## D.6 A (PARTIAL) PARAMETERIZATION OF THE ANTI-DE SITTER SPACE

**Proposition D.6.1.** *Let $p, q \geq 0$ and $d := p + q$. Consider the following space:*

$$\mathbb{A}^{p+1,q} := \mathbb{R}_{>0} \times \mathbb{R}^{p,q}, \tag{274}$$

*which we endow with the metric $\eta^{\mathbb{A}^{p+1,q}}$, which is given at $x \in \mathbb{A}^{p+1,q}$ and with $v_1, v_2 \in \mathrm{T}_x\mathbb{A}^{p+1,q} = \mathbb{R}^{p+1,q}$ as follows:*

$$\eta_x^{\mathbb{A}^{p+1,q}}(v_1, v_2) := \frac{\eta^{p+1,q}(v_1, v_2)}{|x^0|^2}. \tag{275}$$

*Then consider the map:*

$$\phi : \mathbb{A}^{p+1,q} \to \mathbb{R}^{p+1,q+1}, \quad x \mapsto \phi(x) =: y, \tag{276}$$

$$\tag{277}$$

$$\phi^0(x) := \frac{1 - \eta^{p+1,q}(x, x)}{2x^0} = \frac{1}{2x^0} - \frac{x^0}{2} - \frac{1}{2x^0}\eta^{p,q}(x^{1:d}, x^{1:d}), \tag{278}$$

$$\phi^{1:d}(x) := \frac{x^{1:d}}{x^0}, \tag{279}$$

$$\phi^{d+1}(x) := \frac{1 + \eta^{p+1,q}(x, x)}{2x^0} = \frac{1}{2x^0} + \frac{x^0}{2} + \frac{1}{2x^0}\eta^{p,q}(x^{1:d}, x^{1:d}). \tag{280}$$

*Then $\phi$ is a well-defined isometric embedding and its range coincides with the subspace:*

$$\widetilde{\mathrm{AdS}}^{p+1,q} := \left\{ y \in \mathbb{R}^{p+1,q+1} \mid \eta^{p+1,q+1}(y, y) = -1 \ \wedge \ y^0 + y^{d+1} > 0 \right\}. \tag{281}$$

*An inverse $\phi^{-1} : \widetilde{\mathrm{AdS}}^{p+1,q} \to \mathbb{A}^{p+1,q}$ of $\phi$ is given by:*

$$x^0 := (\phi^{-1})^0(y) = \frac{1}{y^0 + y^{d+1}} > 0, \tag{282}$$

$$x^{1:d} := (\phi^{-1})^{1:d}(y) = \frac{y^{1:d}}{y^0 + y^{d+1}}. \tag{283}$$

*Proof.* We first show that $\phi$ is well-defined. For this we compute for $y := \phi(x)$:

$$4|x^0|^2 \cdot \eta^{p+1,q+1}(y, y) = \left(1 - \eta^{p+1,q}(x, x)\right)^2 + 4\eta^{p,q}(x^{1:d}, x^{1:d}) - \left(1 + \eta^{p+1,q}(x, x)\right)^2 \tag{284}$$

$$= -4\eta^{p+1,q}(x, x) + 4\eta^{p,q}(x^{1:d}, x^{1:d}) \tag{285}$$

$$= -4|x^0|^2, \tag{286}$$

implying: $\eta^{p+1,q+1}(y, y) = -1$. Furthermore:

$$y^0 + y^{d+1} = \frac{1 - \eta^{p+1,q}(x, x)}{2x^0} + \frac{1 + \eta^{p+1,q}(x, x)}{2x^0} = \frac{1}{x^0} > 0. \tag{287}$$

Together, this shows: $\phi(x) = y \in \widetilde{\mathrm{AdS}}^{p+1,q} \subseteq \mathbb{R}^{p+1,q+1}$. It is easy to see that $\phi^{-1}$ is the left-inverse to $\phi$, i.e.:

$$\phi^{-1} \circ \phi(x) = x \in \mathbb{A}^{p+1,q}. \tag{288}$$

To show that $\phi^{-1}$ is also a right-invers let $y \in \widetilde{\text{AdS}}^{p+1,q}$ and $x := \phi^{-1}(y)$. Then compute:

$$\phi^{1:d}(x) = \frac{1}{x^0} x^{1:d} = \left(y^0 + y^{d+1}\right) \cdot \frac{y^{1:d}}{y^0 + y^{d+1}} = y^{1:d}, \tag{289}$$

$$\phi^0(x) = \frac{1}{2x^0} - \frac{x^0}{2} - \frac{1}{2x^0} \eta^{p,q}(x^{1:d}, x^{1:d}) \tag{290}$$

$$= \frac{y^0 + y^{d+1}}{2} - \frac{1}{2\left(y^0 + y^{d+1}\right)} - \frac{1}{2\left(y^0 + y^{d+1}\right)} \eta^{p,q}(y^{1:d}, y^{1:d}) \tag{291}$$

$$= \frac{1}{2}\left(\left(y^0 + y^{d+1}\right) - \frac{1}{y^0 + y^{d+1}}\left(1 + \eta^{p,q}(y^{1:d}, y^{1:d})\right)\right) \tag{292}$$

$$= \frac{1}{2}\left(\left(y^0 + y^{d+1}\right) - \frac{1}{y^0 + y^{d+1}}\left(|y^{d+1}|^2 - |y^0|^2\right)\right) \tag{293}$$

$$= \frac{1}{2}\left(\left(y^0 + y^{d+1}\right) - \left(y^{d+1} - y^0\right)\right) \tag{294}$$

$$= y^0. \tag{295}$$

Similarly:

$$\phi^{d+1}(x) = \frac{1}{2x^0} + \frac{x^0}{2} + \frac{1}{2x^0} \eta^{p,q}(x^{1:d}, x^{1:d}) \tag{296}$$

$$= \frac{y^0 + y^{d+1}}{2} + \frac{1}{2\left(y^0 + y^{d+1}\right)} + \frac{1}{2\left(y^0 + y^{d+1}\right)} \eta^{p,q}(y^{1:d}, y^{1:d}) \tag{297}$$

$$= \frac{1}{2}\left(\left(y^0 + y^{d+1}\right) + \frac{1}{y^0 + y^{d+1}}\left(1 + \eta^{p,q}(y^{1:d}, y^{1:d})\right)\right) \tag{298}$$

$$= \frac{1}{2}\left(\left(y^0 + y^{d+1}\right) + \frac{1}{y^0 + y^{d+1}}\left(|y^{d+1}|^2 - |y^0|^2\right)\right) \tag{299}$$

$$= \frac{1}{2}\left(\left(y^0 + y^{d+1}\right) + \left(y^{d+1} - y^0\right)\right) \tag{300}$$

$$= y^{d+1}. \tag{301}$$

This shows that:

$$\phi \circ \phi^{-1}(y) = y \in \widetilde{\text{AdS}}^{p+1,q}. \tag{302}$$

We now compute the differential of $\phi$:

$$d\phi_x = \begin{bmatrix} -\frac{1}{2|x^0|^2} - \frac{1}{2} + \frac{\eta^{p,q}(x^{1:d}, x^{1:d})}{2|x^0|^2} & -\frac{1}{x^0}(x^{1:d})^\top \Delta^{p,q} \\ -\frac{x^{1:d}}{|x^0|^2} & \frac{I_d}{x^0} \\ -\frac{1}{2|x^0|^2} + \frac{1}{2} - \frac{\eta^{p,q}(x^{1:d}, x^{1:d})}{2|x^0|^2} & \frac{1}{x^0}(x^{1:d})^\top \Delta^{p,q} \end{bmatrix} \tag{303}$$

and the metric at $x \in \mathbb{A}^{p+1,q}$:

$$(d\phi_x)^\top \Delta^{p+1,q+1} d\phi_x = (d\phi_x)^\top \begin{bmatrix} 1 & 0 & 0 \\ 0 & \Delta^{p,q} & 0 \\ 0 & 0 & -1 \end{bmatrix} d\phi_x. \tag{304}$$

We compute the entries of the last matrix separately. First, top left entry:

$$\frac{1}{4}\left(1 - \frac{\eta^{p,q}(x^{1:d}, x^{1:d})}{|x^0|^2} + \frac{1}{|x^0|^2}\right)^2 + \frac{\eta^{p,q}(x^{1:d}, x^{1:d})}{|x^0|^4} - \frac{1}{4}\left(1 - \frac{\eta^{p,q}(x^{1:d}, x^{1:d})}{|x^0|^2} - \frac{1}{|x^0|^2}\right)^2 \tag{305}$$

$$= \frac{1}{|x^0|^2}\left(1 - \frac{\eta^{p,q}(x^{1:d}, x^{1:d})}{|x^0|^2}\right) + \frac{\eta^{p,q}(x^{1:d}, x^{1:d})}{|x^0|^4} \tag{306}$$

$$= \frac{1}{|x^0|^2}. \tag{307}$$

Note that in the middle we used the formula: $(a+b)^2 - (a-b)^2 = 4ab$. Next, top right entry:

$$\frac{1}{2}\left(\left(1 - \frac{\eta^{p,q}(x^{1:d}, x^{1:d})}{|x^0|^2}\right) + \frac{1}{|x^0|^2}\right)\frac{(x^{1:p})^\top \Delta^{p,q}}{x^0} - \frac{(x^{1:d})^\top \Delta^{p,q}}{|x^0|^3} \tag{308}$$

$$-\frac{1}{2}\left(\left(1 - \frac{\eta^{p,q}(x^{1:d}, x^{1:d})}{|x^0|^2}\right) - \frac{1}{|x^0|^2}\right)\frac{(x^{1:d})^\top \Delta^{p,q}}{x^0} \tag{309}$$

$$= \frac{1}{2}\left(\frac{2}{|x^0|^2}\right)\frac{(x^{1:d})^\top \Delta^{p,q}}{x^0} - \frac{(x^{1:d})^\top \Delta^{p,q}}{|x^0|^3} \tag{310}$$

$$= 0. \tag{311}$$

By symmetry also the bottom left equals 0. Finally, bottom right entry:

$$\frac{\Delta^{p,q}x^{1:d}}{x^0}\frac{(x^{1:d})^\top \Delta^{p,q}}{x^0} + \frac{I_p}{x^0}\Delta^{p,q}\frac{I_p}{x^0} - \frac{\Delta^{p,q}x^{1:d}}{x^0}\frac{(x^{1:d})^\top \Delta^{p,q}}{x^0} = \frac{\Delta^{p,q}}{|x^0|^2}. \tag{312}$$

Together we get:

$$(d\phi_x)^\top \Delta^{p+1,q+1} d\phi_x = \frac{1}{|x^0|^2}\begin{bmatrix} 1 & 0 \\ 0 & \Delta^{p,q} \end{bmatrix} = \frac{1}{|x^0|^2}\Delta^{p+1,q}. \tag{313}$$

This shows the claim. $\qquad\qquad\square$

**Theorem D.6.2.** *The map:*

$$\mathbb{A}^{p+1,q} \cong \widetilde{\mathrm{AdS}}^{p+1,q} \to \mathrm{AdS}^{p+1,q}, \qquad\qquad x \mapsto \phi(x) =: y \mapsto [y], \tag{314}$$

*is an isometric embedding. The complement of its range is given by the closed subspace:*

$$\mathcal{Z} := \left\{[z] \in \mathrm{AdS}^{p+1,q} \,\big|\, z^0 + z^{d+1} = 0\right\}. \tag{315}$$

*The left-inverse for $[z] \in \mathrm{AdS}^{p+1,q}$ (with $z^0 + z^{d+1} \neq 0$) is given by:*

$$y := \frac{\mathrm{sgn}(z^0 + z^{d+1})}{\sqrt{|\eta^{p+1,q+1}(z,z)|}} \cdot z \in \widetilde{\mathrm{AdS}}^{p+1,q}, \tag{316}$$

*and to get $x \in \mathbb{A}^{p+1,q}$ we put:*

$$x^0 := (\phi^{-1})^0(y) = \frac{\sqrt{|\eta^{p+1,q+1}(z,z)|}}{|z^{d+1} + z^0|} > 0, \tag{317}$$

$$x^{1:d} := (\phi^{-1})^{1:d}(y) = \frac{z^{1:d}}{z^{d+1} + z^0}. \tag{318}$$

**Remark D.6.3.** *Note that in Theorem D.6.2 in the Euclidean case, i.e. if $q = 0$, the subspace $\mathcal{Z}$ is empty, as $z^0 + z^{d+1} = 0$ would imply:*

$$0 > \eta^{p+1,q+1}(z,z) \tag{319}$$

$$= |z^0|^2 + \eta^{p,q}(z^{1:d}, z^{1:d}) - |z^{d+1}|^2 \tag{320}$$

$$= (z^0 + z^{d+1})(z^0 - z^{d+1}) + \eta^{p,q}(z^{1:d}, z^{1:d}) \tag{321}$$

$$= \eta^{p,q}(z^{1:d}, z^{1:d}), \tag{322}$$

*which is not possible for $q = 0$. It follows that for $q = 0$ we get isometries:*

$$\mathbb{A}^{p+1,0} \cong \widetilde{\mathrm{AdS}}^{p+1,0} \cong \mathrm{AdS}^{p+1,0}. \tag{323}$$

**Remark D.6.4.** *For $p, q \geq 0$, $d := p + q$, consider the map:*

$$\bar{\phi} : \mathbb{R}_{\geq 0} \times \mathbb{R}^{p,q} \to \mathbb{P}^{d+1}, \qquad x \mapsto [1 - \eta^{p+1,q}(x,x) : 2x^{1:d} : 1 + \eta^{p+1,q}(x,x)]. \tag{324}$$

*Then $\bar{\phi}$ is well-defined. For $x^0 = 0$ the map $\bar{\phi}$ coincides with the conformal embedding $\iota$ of $\mathbb{R}^{p,q}$ into the conformal compactification $\mathbb{M}^{p,q} \subseteq \mathbb{P}^{d+1}$ of $\mathbb{R}^{p,q}$, and, for $x^0 > 0$, $\bar{\phi}$ coincides with the isometric embedding $\phi$ of $\mathbb{A}^{p+1,q}$ into $\mathrm{AdS}^{p+1,q} \subseteq \mathbb{P}^{d+1}$.*

### D.7 THE EUCLIDEAN CASE ($p \geq 2$ AND $q = 0$)

In the following we now consider the Euclidean case, i.e. $q = 0$, for $p \geq 2$, $d = p$.

**Remark D.7.1.** *Note that we have:*

$$\mathbb{S}^0 = \{\pm 1\}, \qquad\qquad \mathbb{S}^{d,0} = \mathbb{S}^d \times \{\pm 1\}. \tag{325}$$

*So, for $q = 0$, This thus simplifies the description of the conformal compactification, the corresponding de Sitter and anti-de Sitter space.*

**Corollary D.7.2** (The conformal compactification of the Euclidean space)**.** *We have the isometry for $d \geq 0$ and $q = 0$, $p = d$:*

$$\mathbb{M}^{d,0} \cong \mathbb{S}^d \times \{+1\} \cong \mathbb{S}^d, \tag{326}$$

$$[z] \mapsto \left( \frac{z^{0:d}}{z^{d+1}}, +1 \right) \mapsto \frac{z^{0:d}}{z^{d+1}}. \tag{327}$$

*The conformal embedding is then given as:*

$$\tau_+ : \mathbb{R}^{d,0} \to \mathbb{M}^{d,0} \cong \mathbb{S}^d \subseteq \mathbb{R}^{d+1}, \qquad \tau_+(x) := \left( \frac{1 - \eta^{d,0}(x,x)}{1 + \eta^{d,0}(x,x)}, \frac{2x}{1 + \eta^{d,0}(x,x)} \right), \tag{328}$$

*which coincides with the invers stereographic projection (from the "south pole").*

*The group $\mathrm{PO}(d+1,1)$ then acts on $\mathbb{S}^d$ as follows, $[\Lambda] \in \mathrm{PO}(d+1,1)$, $y = y^{0:d} \in \mathbb{S}^d$:*

$$[\Lambda].y = \frac{\left( \Lambda \begin{bmatrix} y \\ 1 \end{bmatrix} \right)^{0:d}}{\left( \Lambda \begin{bmatrix} y \\ 1 \end{bmatrix} \right)^{d+1}}, \tag{329}$$

*where $\Lambda \begin{bmatrix} y \\ 1 \end{bmatrix}$ denotes the matrix vector product of $(d+2) \times (d+2)$-matrix $\Lambda$ and the vector $y$ with a 1 appended as the last component. Note that the indices of the components of $y \in \mathbb{S}^d \subseteq \mathbb{R}^{d+1}$ range from $i = 0, 1, \ldots, d$.*

**Remark D.7.3.** *Note that we always have:*

$$\mathrm{Isom}(\mathbb{S}^d) \cong \mathrm{O}(d+1), \tag{330}$$

*via the usual action and for $d \geq 2$:*

$$\mathrm{ConfDiff}(\mathbb{S}^d) \cong \mathrm{PO}(d+1,1), \tag{331}$$

*via the action above. The latter is part of Theorem D.4.2.*

**Remark D.7.4.** *Recall that by Proposition D.6.1, Theorem D.6.2 and Remark D.6.3 for $q = 0$ we have isometries:*

$$\mathbb{A}^{d+1,0} \cong \widetilde{\mathrm{AdS}}^{d+1,0} \cong \mathrm{AdS}^{d+1,0}, \tag{332}$$

*where we can slighly re-write[16] $\widetilde{\mathrm{AdS}}^{d+1,0}$ as:*

$$\widetilde{\mathrm{AdS}}^{d+1,0} = \left\{ y \in \mathbb{R}^{d+1,1} \,\big|\, \eta^{d+1,1}(y,y) = -1, y^{d+1} > 0 \right\}. \tag{333}$$

*With this we get the isometry:*

$$\mathrm{AdS}^{d+1,0} \cong \widetilde{\mathrm{AdS}}^{d+1,0} \subseteq \mathbb{R}^{d+1,1}, \qquad\qquad [z] \mapsto \frac{\mathrm{sgn}(z^{d+1})}{\sqrt{|\eta^{d+1,1}(z,z)|}} \cdot z =: y. \tag{334}$$

*Note that then:*

$$[y] = [z] \in \mathrm{AdS}^{d+1} \subseteq \mathbb{P}^{d+1}. \tag{335}$$

---

[16]Note that for $y \in \mathbb{R}^{d+1,1}$ with $\eta^{d+1,1}(y,y) < 0$ we have: $|y^{d+1}|^2 > |y^0|^2$. So, for those $y$, we have the equivalence: $y^{d+1} > 0 \iff y^0 + y^{d+1} > 0$.

*Furthermore, for $[\Lambda] \in \mathrm{PO}(d+1, 1)$ we define the action on $y \in \widetilde{\mathrm{AdS}}^{d+1,0}$:*

$$[\Lambda].y = \mathrm{sgn}((\Lambda y)^{d+1}) \cdot (\Lambda y), \tag{336}$$

*where $\Lambda y$ denotes the usual matrix product. Note that this is well-defined and that:*

$$[\Lambda].y \in \widetilde{\mathrm{AdS}}^{d+1,0}, \qquad\qquad [[\Lambda].y] = [\Lambda y] = [\Lambda][y] \in \mathrm{AdS}^{d+1,0}. \tag{337}$$

*This shows that the above isometry $\widetilde{\mathrm{AdS}}^{d+1,0} \cong \mathrm{AdS}^{d+1,0}$ is also $\mathrm{PO}(d+1, 1)$-equivariant.*

**Definition/Lemma D.7.5** (Geodesic distance of $\mathrm{AdS}^{d+1,0}$)**.** *The geodesic distance on $\widetilde{\mathrm{AdS}}^{d+1,0}$, $\mathrm{AdS}^{d+1,0}$ and $\mathbb{A}^{d+1,0}$, resp., is given by:*

$$d^{\widetilde{\mathrm{AdS}}^{d+1,0}}(y_1, y_2) := \mathrm{arccosh}\left(|\eta^{d+1,1}(y_1, y_2)|\right), \tag{338}$$

$$d^{\mathrm{AdS}^{d+1,0}}([z_1], [z_2]) := \mathrm{arccosh}\left(\frac{|\eta^{d+1,1}(z_1, z_2)|}{\sqrt{|\eta^{d+1,1}(z_1, z_1)| \cdot |\eta^{d+1,1}(z_2, z_2)|}}\right), \tag{339}$$

$$d^{\mathbb{A}^{d+1,0}}(x_1, x_2) := \mathrm{arccosh}\left(\frac{|x_1^0|^2 + |x_2^0|^2 + \eta^{d,0}(x_1^{1:d}, x_2^{1:d})}{2x_1^0 x_2^0}\right). \tag{340}$$

