# OpenReview forum: "AdS-GNN - a Conformally Equivariant Graph Neural Network"
_ICLR.cc/2026/Conference — ICLR 2026 Poster_

### Official Review · Reviewer_PdJG · 2025-10-30

**Soundness:** 4
**Presentation:** 3
**Contribution:** 3
**Rating:** 8
**Confidence:** 4

**Summary:**

This paper introduces a new architecture, AdS-GNN, which is equivariant under conformal symmetries (preserving angles).
Conformal symmetries include scaling and a type of inversion, called the special conformal transformation, in addition to rotation and translations.
The key idea is to first map the data onto an Anti-de Sitter space, a uniform negatively curved space. The reason for this mapping is that at its boundary, AdS has conformal symmetries. The mapping to AdS far from its boundary breaks invariance under the special conformal transformations, but the rest remain intact. This mapping is motivated by the so-called AdS/CFT correspondence.
The approximate conformal inviariance on AdS corresponds to a certain norm being invariant and that norm is used to define equivariant and invariant features for the GNN.

They apply AdS-GNN to a few problems and show strong results in physics examples, especially those enjoying some scale invariance.

**Strengths:**

1. AdS-GNN outperforms other baseline on the physics tasks, such as Ising and N-body electrostatic simulation
2. It performs well even in small data regimes.
3. The idea of first mapping to AdS before applying the model is interesting and inspired by known fact from AdS/CFT.
4. This is a nice model for equivariant model scale-invariant systems.
5. The mapping to AdS is fairly easy, though it involves some preprocessing. Other than that, the model seems easy to implement.

**Weaknesses:**

1. Given the fact that special conformal is broken by the AdS bulk embedding, what is left is scaling + rotation + translation. So, really the model is adding scale equivariance to EGNN. This should be discussed or emphasized.
2. The choice of experiments isn't explained much. As above, I think AdS-GNN works best for scale-invariant problems. In physics, these are the systems which we expect to have conformal invariance, as discussed in the intro. This may be why on Superpixel MNIST they don't outperform EGNN.
3. A general discussion of limitations and realm of applicability is needed.
4. All systems and datasets used in experiments seem very small. It is fine as a proof of concept, but scalability needs to be discussed.

**Questions:**

1. How does the run time, including AdS embedding, scale with size of the systems?
2. Do you only use scalars and vectors as features in AdS-GNN? This should work fine for Ising which has i=only quadratic interactions. But what if you had a spin-glass with higher order interactions? Do you think higher tensors would be needed to learn correlations there? Is it known which of those systems, or in what regime they would be scale-invariant?
3. Any good criteria for deciding when to use AdS-GNN, other than obvious scale-invariance?
4. Some of the scale-invariant systems here seem fully conformal invariant. How much does the breaking of special conformal cause a problem? How do you measure the effect of this breaking?

---

> ### Author Response · Authors · 2025-11-21
> **Rebuttal by Authors**
>
> We thank the reviewer for their encouraging feedback and careful reading. We are grateful for acknowledging the strengths of the proposed approach, and we hope to clarify remaining questions in our response.
>
> # **Question 1 (computational cost)**:
> The runtime of the model itself is linear w.r.t. the model size. The embedding, on the other hand, depends on whether the input point cloud is associated with a graph/mesh that can be used for lifting. If that is the case, then the cost of embedding is linear as well. If there is no graph given, it has to be built for the lifting and the cost of embedding therefore depends on the algorithm of graph construction. In our experiments, we used k-nearest neighbours, which is O(NlogN), although one can as well use a radius graph, bringing the cost to linear.
>
> **Action taken**: we added a paragraph discussing the computational cost of the approach at the end of Section 4.
>
> # **Question 2 (representations of conformal algebra, connection to other models)**:
> a) This is a great question. We test two models: one of them, AdS-GNN (defined in Eq (15)) uses only scalars. The other, AdS-CEGNN, (defined in Eq (16)) uses features that are elements of the Clifford algebra of O(d+1,1) and so includes vectors, bivectors, multivectors, etc.
>
> b) The question as to what representations are needed as features is very interesting. Even in the case of Ising we are not certain that it is obvious that scalar features should be sufficient. To elaborate on this point: we note that even if the microscopic Ising model has only quadratic spin-interactions, at the critical point of the model we have a nontrivial interacting system which at long distances is described by the Ising conformal field theory (CFT). This CFT typically has operators transforming under various representations of the conformal group. In the case of the 3d Ising CFT there are many operators of higher spin (e.g. the so-called “stress-energy tensor”, present in every CFT, has spin 2) which one might have thought would be needed to be included as latent variables. (Indeed in AdS/CFT, from which this work draws inspiration, it is important to include such operators as intermediate states in Feynman diagrams [1], which are perhaps the physics analogue of latent variables)
>
> c) Nevertheless the completely scalar model AdS-GNN reproduces the Ising correlators very well even without these higher spin features. It would be interesting to carefully dissect the construction to understand how it manages to achieve this. For now however we believe that the scalar AdS-GNN alone would be able to represent any correlation function of scalar operators in a conformal field theory, independent of whether the microscopic description has higher-order interactions or not.
>
> d) Re: spin glasses – when referring to “higher order interactions” we believe the referee is discussing e.g. the p-spherical spin glass [2], which has an all-to-all coupling of all N spins. In such a model spatial locality is lost and so it is not really conformal in a useful sense. However there are other models without higher order interactions but with the main features of spin glasses (i.e. disordered couplings) and only local quadratic couplings of the spins, see. e.g. [3]. In such models there is indeed a conformal fixed point which could be studied with our methods.
>
> # **Question 3 (when is AdS-GNN useful?)**
> We believe that AdS-GNN is likely to be useful whenever there are long-range interactions present in the system. This is however very related to scale-invariance; any interaction that is *not* scale-invariant is likely to die away over sufficiently long scales.
> Indeed, the examples drawn from physics are fully conformally invariant. It does not appear that the breaking of special conformal invariance really causes any issues here at all, as the performance is excellent, much better than that of the baselines. It seems possible that even if the model is not constrained to be perfectly SCT-invariant, it approximately learns to be so from the data in much the same way that a non-equivariant model can approximately learn a symmetry from data.
>
> # **Question 4 (SCT breaking)**
> In the application to SuperPixelMNIST we measured the effect of the breaking empirically by performing a special conformal transformation on the test set and observing how the accuracy of the model degraded, shown in Figure 3: we observed the decline to be quite mild.
>
> References:
> 1. d'Hoker, E., Freedman, D.Z., Mathur, S.D., Matusis, A. and Rastelli, L., 1999. Graviton exchange and complete four-point functions in the AdS/CFT correspondence. Nuclear Physics B, 562(1-2), pp.353-394
> 2. Crisanti, A., Sommers, H.J. The spherical p-spin interaction spin glass model: the statics. Z. Physik B - Condensed Matter 87, 341–354 (1992). https://doi.org/10.1007/BF01309287
>
> 3. Zohar Komargodski and David Simmons-Duffin 2017 J. Phys. A: Math. Theor. 50 154001

---

### Official Review · Reviewer_jAnW · 2025-10-31

**Soundness:** 3
**Presentation:** 3
**Contribution:** 3
**Rating:** 6
**Confidence:** 3

**Summary:**

The paper presents a conformally equivariant graph neural network through showing how to lift data from Euclidean space to Anti de Sitter (AdS) space. Distance metrics in AdS space can then be used in conjunction with invariant message passing to create neural networks that are equivariant to the conformal group. Experimental results on computer vision tasks (SuperPixel MNIST and shape segmentation) and physics tasks (2d/3d Ising model and N-body simulation) are presented. In particular, for the physics tasks, AdS-GNN seems to require less data and generalizes better. Additionally, it recovers the correct conformal dimension, illustrating interpretability.

**Strengths:**

The paper presents an interesting new framework for conformal group equivariance, using ideas from theoretical physics (and building open pre-existing work for constructing group equivariant convolutional layers). It is well-organized with experiments in multiple sub-domains, and provides a self contained introduction to the conformal group. I particularly like the figure at the top of pg. 3 showing possible transformations under the conformal group. Experimental results show that AdS-GNN maintains scale invariance (e.g. SuperPixel MNIST). The physics tasks are good illustrations of the utility of AdS-GNN, showing that AdS-GNN outperforms other models (EGNN and MPNN) for predicting N-point correlation functions for the 2D Ising model and requires less training data. I found it particularly interesting as well that the learned values of the conformal dimensions match the ground truth, and it would be interesting to explore this point further in more complex physics datasets.

**Weaknesses:**

I am not sure of the usefulness of the image experiments. I wouldn’t expect image datasets to require conformal equivariance/to me the orientation of the image would probably matter the most. However, the AdS-GNN model is invariant, so this orientation is not taken into account. It does not seem to be persuasive that one would use AdS-GNN in a computer vision setting.

I found the part about embedding points in AdS somewhat confusing, and the choice of regular $z_0$ is unclear. I am concerned that this would impact the dataset/break certain symmetries (see questions).  I think further clarification is needed in this section, perhaps an additional figure showing another embedded shape (or embeddings of circles, triangles as in the shape analysis data) would be helpful.

AdS is unable to handle orientation and relies on invariant descriptors. This seems to be a significant limitation, for both image datasets and physics datasets, in light of other work on equivariant neural networks with message passing of higher-order tensorial features (e.g. [1]).

The shape segmentation task is quite toy, the authors could consider exploring a more realistic dataset such as ModelNet.

Overall, most of the experiments seem to test scale invariance, but there are pre-existing scale invariant models (from my understanding). It may be good to benchmark against these pre-existing scale invariant models and see what conformal invariance actually gives us.

**Questions:**

What is the computational cost of these models compared to other invariant models (e.g. EGNN)? Does the lifting procedure incur significant computational costs?

Another scientific domain of interest could be fluid dynamics. At sufficiently high Reynolds number, the statistics of turbulent motions in the so-called “inertial range” become universal and exhibit scale invariance properties [2]. I would be interested to see the performance of these models on turbulence modeling tasks.

In what settings would conformal equivariance be useful for images? It seems like it would be more useful in dynamical systems/more-physics motivated problems as shown in the physics task experiments. Is there further motivation for why one would want robustness to conformal transformations in computer vision tasks?

Would it be possible to have non-scalar features in message passing? Were there any experiments done extending the model to include non-scalar features?

How is the regulator $z_0$ chosen, and how does this impact the resulting lifting procedure? Are there other ways that one could lift data into AdS/why was this way in particular chosen? If the lifting procedure breaks certain conformal transformations, which transformations does it break? Figure 3 shows one broken transformation, but what do these “special” transformations correspond to physically, and why would this not be a problem that these are broken? Is it accurate to call the model conformally equivariant if this is the case (maybe approximately conformally equivariant)? Correct me if I'm wrong, but currently it seems conformally invariant rather than equivariant?

Were any comparisons done to pre-existing scale invariant models? This would be interesting to include with comparison to the rotationally/translationally invariant models.

[1] Batatia et al. MACE: Higher Order Equivariant Message Passing Neural Networks for Fast and Accurate Force Fields (2022).

[2] Pope, Stephen. Turbulent Flows. (Cambridge University Press, 2000).

---

> ### Author Response · Authors · 2025-11-21
> **Rebuttal by Authors (part 1/2)**
>
> We are very grateful to the referee for their careful reading and positive assessment of our submission. We agree with the referee that one of the most interesting aspects of our construction is the ability to extract and interpret conformal dimensions. We respond to the detailed and insightful questions below:
>
> ## **Question 1 (computational cost)**:
> That is an excellent question. The computational cost of our approach decomposes into two components: the AdS embedding procedure (Algorithm 1) and applying the graph neural network. In the former, we employ $k$-nearest neighbors which has $\mathcal{O}(N \log N)$ complexity. An MPNN scales linearly with the number of nodes $N$ in the graph. We note that in the case when a graph is available, one potentially can use it during the lifting, thus alleviating the overhead of $k$-nearest neighbors.
>
> **Action taken**: we added a paragraph discussing the computational cost of the approach at the end of Section 4.
>
> ## **Question 2 (Fluid dynamics)**:
> We completely agree with the referee that this is a very interesting task which is indeed on our radar. The situation may be even better, as there are some indications in the literature that the inertial range has nontrivial conformal invariance and not scale invariance [1]. For the scope of the paper, we focused on the different set of experiments, and left the task for future work.
>
> ## **Question 3 (Application to images)**:
> The advantages of the proposed approach for image-related tasks are following:
> - It consistently handles size variations (objects at different distances, zoom levels, etc.), while rotation equivariance handles orientation changes, which is only complementary.
> - The model is consistent with local scale variations that are common in real-world vision - objects appear at different sizes due to distance from camera, intrinsic size differences, etc.
> - For robotics and computer vision, this means the network maintains consistent predictions when objects appear at varying scales within scenes, potentially improving robustness without requiring extensive multi-scale training data or test-time augmentation.
>
> **Action taken**: we added a paragraph discussing possible application beyond physics to the conclusion.
>
> ## **Question 4 (Non-scalar features)**:
> We test two models: one of them, AdS-GNN (defined in Eq (15)) uses only scalars. The other, AdS-CEGNN, (defined in Eq (16)) uses features that are elements of the Clifford algebra of O(d+1,1) and so includes vectors, bivectors, multivectors, etc. In the original submission we benchmarked AdS-CEGNN on the N-body particle motion task (where it is SOTA).

---

> ### Author Response · Authors · 2025-11-21
> **Rebuttal by Authors (part 2/2)**
>
> ## **Question 5 (The choice of $z_0$ & lifting & SCT)**:
> We discuss the regulator, uniqueness of lifting, and the breaking of special conformal transformations separately.
> - The regulator does not appear to be very important if it is taken small, as the limit is smooth. In fact in Appendix A3 we discuss a more abstract approach where the regulator is zero from the beginning: in this slightly more abstract construction the points are originally embedded on the null cone tangent to the hyperboloid given by AdS, but it coincides with the approach used in the bulk of the paper in the limit where the regulator vanishes. We added another equation to the Appendix to make this more explicit.
>
> - We chose this particular method of lifting by an analogy with the AdS/CFT correspondence in quantum gravity: in AdS/CFT it is well-known that objects associated with larger “sizes” are deeper in the interior of AdS (see e.g. [2]). This lifting approach directly captures that intuition, where the “size” in question is the (appropriately averaged) distance of a given point from its k nearest neighbours. There could be other ways to do the lifting, and this is an interesting question to explore.
>
> - Physically, a special conformal transformation is a (maybe somewhat unfamiliar) transformation that preserves angles, and is mathematically given by the rightmost column in Eq (4). It is a bona-fide symmetry of many physical systems (e.g. the 2d Ising model at a critical point, Maxwell electrodynamics, etc.).
>
> - The lifting exactly preserves the subgroup of the conformal group generated by translations, rotations, and scaling, but not special conformal transformations. This breaking is somewhat expected as many calculations in physics also require these to be broken. Empirically however we find the breaking to be rather mild, as we show in the second panel in Figure 3. Also it does not seem to hurt the model from obtaining good performance in various tasks. It is possible that though the special conformal equivariance is not enforced by the architecture, it managed to learns it from data; this would be interesting to investigate further.
>
> - We stress that every step after the lifting (i.e. the manipulations in AdS) are equivariant under the full conformal group, which (together with brevity) motivated the name. In the case of AdS-GNN with only scalar features they are indeed invariant as the referee suggests; in the case of AdS-CEGNN they are equivariant, as the features transform in nontrivial representations.
>
> **Action taken**: we added a new equation (30) to Appendix A.3 to explain the regulator-free approach to lifting.
>
> ## **Question 6 (scale-invariant point cloud methods)**:
> At the time of writing we were not actually aware of comparable scale-invariant methods operating on point clouds; there is a much larger literature acting on images.
>
> References:
> 1. Bernard, Denis, Guido Boffetta, Antonio Celani, and Gregory Falkovich. "Conformal invariance in two-dimensional turbulence." Nature Physics 2, no. 2 (2006): 124-128
> 2. Susskind, L. and Witten, E., 1998. The holographic bound in anti-de Sitter space. arXiv preprint hep-th/9805114.

---

> > ### Comment · Reviewer_jAnW · 2025-11-27
> > **Thanks for the responses**
> >
> > Thanks for the detailed responses to my questions! I agree that it would be interesting to see if the special conformal equivariance can be learned from the data. I did some digging as well, and it seems that there are not many scale-invariant models currently built for point clouds. I feel that this is novel and useful work and my concerns have been addressed. I am raising my score to 8/recommending the paper be accepted.

---

> > > ### Author Response · Authors · 2025-11-27
> > > **Official Comment by Authors**
> > >
> > > Thank you very much for your positive feedback. Your suggestions have significantly improved our work. We will continue refining the paper based on the other reviewers' comments. We truly appreciate your support.

---

### Official Review · Reviewer_HgBi · 2025-11-01

**Soundness:** 2
**Presentation:** 3
**Contribution:** 2
**Rating:** 4
**Confidence:** 3

**Summary:**

This paper proposes "AdS-GNN," a new graph neural network that is equivariant to conformal transformations (translations, rotations, scale transformations, and special conformal transformations). To achieve this equivariance, the authors, inspired by insights from the AdS/CFT correspondence in physics, introduce a method to lift input data from flat Euclidean space $\mathbb{R}^d$ to Anti-de Sitter (AdS) space $AdS_{d+1}$, which has one additional dimension. On AdS space, the conformal transformations of the original space manifest as isometric transformations (distance-preserving transformations). Therefore, the paper efficiently achieves conformal equivariance by constructing a message-passing GNN that utilizes the proper distance of AdS space as an invariant. The proposed method was evaluated on tasks from computer vision (e.g., SuperPixel MNIST) and statistical physics (e.g., 2D/3D Ising models). Particularly in the physics tasks, it was confirmed to show high generalization performance even under extrapolation (OOD) scenarios, such as scale transformations or changes in system size (number of points). Furthermore, the interpretability of the model was also demonstrated, as it was able to extract a physical universal quantity—the scaling dimension of the Ising model—from the trained network with high precision.

**Strengths:**

**Novelty of the Idea**: The application of the profound idea of AdS/CFT correspondence from physics to the context of geometric deep learning, thereby constructing a GNN architecture with conformal equivariance, is highly original and commendable.

**High Affinity with Physics Tasks**: Due to its design background, the proposed method is an excellent fit for tasks in statistical physics where conformal symmetry plays a dominant role (e.g., the Ising model near its critical point).

**Excellent Generalization and Interpretability**: In experiments on physics tasks, AdS-GNN demonstrated performance superior to existing equivariant GNNs (like EGNN). Its robustness to extrapolation tasks, such as changing the system size (number of points), is particularly noteworthy. Furthermore, the fact that the model can automatically learn and extract a physically meaningful universal quantity—the scaling dimension—from data and recover its true value with high precision is a testament to the model's high interpretability and a significant contribution.

**Weaknesses:**

**Lack of Generality and Performance**: The main contribution of this method appears to be limited to physics tasks. As shown by the experimental results (Table 1), in standard image (point cloud) classification tasks like SuperPixel MNIST, the performance does not reach that of existing SOTA methods (e.g., PONITA), and the method's superiority in general-purpose benchmarks has not been demonstrated.

**Limited Applicability**: This method requires prior knowledge that the target data or task possesses "conformal symmetry." Its application to many general machine learning tasks where such strong symmetry does not exist or is unknown is difficult, and the method's utility is inherently restricted.

**Insufficient Appeal to the ICLR Community**: As a result of the above two points, the paper's contribution feels strongly directed primarily at the physics community. There is a lack of discussion or evidence regarding what new possibilities this conformal equivariance approach could bring to the broader ICLR audience (including computer vision, reinforcement learning, natural language processing, etc.).

**Questions:**

**Regarding generality:** While I understand the method's primary contribution lies in physics tasks, what advantages do the authors believe this conformal equivariance approach offers over existing equivariant GNNs (e.g., those with rotational or translational equivariance) in domains outside of physics, such as general computer vision or robotics? I would also like to ask for the authors' insights into why the method failed to achieve SOTA performance on SuperPixel MNIST. Are there specific reasons to consider, such as the approximation error from the lifting procedure, the nature of the point cloud data, or a potential lack of expressive power in the architecture?

---

> ### Author Response · Authors · 2025-11-21
> **Rebuttal by Authors**
>
> We are grateful to the referee for their acknowledgment of the novelty and generalizability of our construction, and for their careful reading and insightful questions. We address them below:
>
> ## **Question 1 (generality)**
>
> This is a great question. The advantages of the proposed approach outside physics-related tasks are following:
> - It consistently handles size variations (objects at different distances, zoom levels, etc.), while rotation equivariance handles orientation changes, which is only complementary.
> - The model is consistent with local scale variations that are common in real-world vision - objects appear at different sizes due to distance from camera, intrinsic size differences, etc.
> - For robotics and computer vision, this means the network maintains consistent predictions when objects appear at varying scales within scenes, potentially improving robustness without requiring extensive multi-scale training data or test-time augmentation.
>
> **Action taken**: we added a paragraph discussing possible application beyond physics to the conclusion.
>
> ## **Question 2 (SuperPixel MNIST performance)**
>
> First, we note that in the submission AdS-GNN obtained an error of 4.07%. Upon further engineering refinements (increasing layers, etc.) we managed to reduce the error to 3% -- it seems possible to close the gap further via scaling the model capacity.
>
> Now regarding the remaining gap: AdS-GNN admits only scalar features, and therefore it indeed lacks expressive power, similar to EGNN and PNITA, in the sense that it is not able to process richer positional/orientational features. On the contrary, the SOTA model, PONITA, does handle such features. That being said, while we believe that achieving good performance is an important dimension to evaluate models on, we also want to stress that robustness and generalization capabilities against various geometric transformations is of importance. Note that none of the previous models were able to perform on conformally transformed (e.g. randomly scaled) inputs at test time, in contrast to our model. Here, for example, is evaluation of the models on randomly scaled and rotated data (accuracy)
> - PONITA: 10%
> - **AdS-GNN (Ours)**: 96.6%

---

> ### Author Response · Authors · 2025-11-27
> **Official Comment by Authors**
>
> Dear Reviewer,
>
> We hope this message finds you well. As the discussion phase closes in the coming days, we wanted to check if our previous responses have addressed your concerns. If you have any remaining questions or suggestions, we would be happy to discuss them further.
>
> We truly appreciate the time and effort you've dedicated to reviewing our work. Thank you again for your thoughtful engagement.
>
> Best regards,
> Authors

---

### Author Response · Authors · 2025-12-03
**Official comment by the authors**

We sincerely thank all reviewers for their time and thoughtful feedback. Based on your valuable suggestions, we have revised the paper to improve its clarity and quality.

We responded directly to each specific question asked. The main modifications to the paper are as follows:

  **Addressing Reviewers HgBi and jAnW**: We have expanded the Conclusion section to include a discussion of potential applications of our approach to computer vision and robotics tasks as part of our future work.

  **Addressing Reviewers jAnW and PdJG**: We have added a dedicated subsection (Section 4.2) discussing the computational cost of our approach.

**Enhanced experimental results**: We have included additional data points in our N-body experiment that demonstrate the extreme data efficiency of our approach. Notably, our method requires only 2-8 input trajectories to outperform the current state-of-the-art approach trained on 10,240 trajectories.

All modifications are highlighted in red text throughout the revised manuscript. We hope these revisions address the reviewers’ concerns.

---

### Meta-Review · Area_Chair_oq62 · 2025-12-20

**Summary:**

The paper introduced a lifting procedure to AdS space that endows EGNNs with equivariance to conformal transformations. Reviewers praised the original idea, writing, and excellent performance on specific tasks. Concerns were raised about limited applications and appeal to general ML audience, and lack of larger-scale real-world experiments.

Initial scores were mostly positive with only HgBi recommending rejection. I believe their specific concerns were addressed in the rebuttal, but some concerns still remain. The authors seemed to respond to each reviewer question but did not specifically address each weaknesses.

Overall I recommend acceptance accepted due to the submission's novelty and potential. The idea introduced was considered interesting and sound and potentially applicable to relevant tasks by the reviewers, even though practical applications at scale were not demonstrated.

**Reviewer Concerns:**

The most serious concerns were raised by HgBi, who recommended rejection mainly due to limited applications to physics and lack of general interest for an ICLR audience. Those were addressed, with the rebuttal highlighting potential applications of local-scale invariance to computer vision and robotics, for example for handling multiple objects of different sizes in the same scene. Although this property was not demonstrated empirically, it does show potential. The rebuttal also demonstrated that the performance on superpixel MNIST improves with hyperparameter tuning, reducing the gap to previous methods.

Some other concerns were not addressed in the rebuttal: 1) jAnW concern about the loss of rotation-equivariance and whether it is worth it, and 2) PdJG request of thorough discussion of limitations and applicability. These reviewers already recommended acceptance so I don't think they would change their minds.

**Reviewer Scores:**

I believe HgBi could raise their score given the clarification about applications to CV and robotics. The other reviewers were already positive and would maintain or increase their scores.

---

### Decision · Program_Chairs · 2026-01-26

Accept (Poster)